# SELF-PLAY WITH EXECUTION FEEDBACK: IMPROVING INSTRUCTION-FOLLOWING CAPABILITIES OF LARGE LANGUAGE MODELS

**Guanting Dong**[*], **Keming Lu, Chengpeng Li**[*]**, Tingyu Xia**[*]**, Bowen Yu**[†]
**Chang Zhou, Jingren Zhou**
Qwen Team, Alibaba Inc.
{dongguanting.dgt,lukeming.lkm,lichengpeng.lcp}@alibaba-inc.com
{xiatingyu.xty, yubowen.ybw,ericzhou.zc,jingren.zhou}@alibaba-inc.com

## ABSTRACT

One core capability of large language models (LLMs) is to follow natural language instructions. However, the issue of automatically constructing high-quality training data to enhance the complex instruction-following abilities of LLMs without manual annotation remains unresolved. In this paper, we introduce AUTOIF, the first scalable and reliable method for automatically generating instruction-following training data. AUTOIF transforms the validation of instruction-following data quality into code verification, requiring LLMs to generate instructions, the corresponding code to verify the correctness of the instruction responses, and unit test samples to cross-validate the code's correctness. Then, execution feedback-based rejection sampling can generate data for Supervised Fine-Tuning (SFT) and Reinforcement Learning from Human Feedback (RLHF) training. AUTOIF achieves significant improvements across three training algorithms, SFT, Offline DPO, and Online DPO, when applied to the advanced open-source LLMs, Qwen2 and LLaMA3, in self-alignment and strong-to-weak distillation settings. Using two widely-used and three challenging general instruction-following benchmarks, we demonstrate that AUTOIF significantly improves LLM performance across a wide range of natural instruction constraints. Notably, AUTOIF is the first to surpass 90% accuracy in IFEval's loose instruction accuracy, without compromising general, math and coding capabilities. Further analysis of quality, scaling, combination, and data efficiency highlights AutoIF's strong generalization and alignment potential. Our code are available at https://github.com/QwenLM/AutoIF

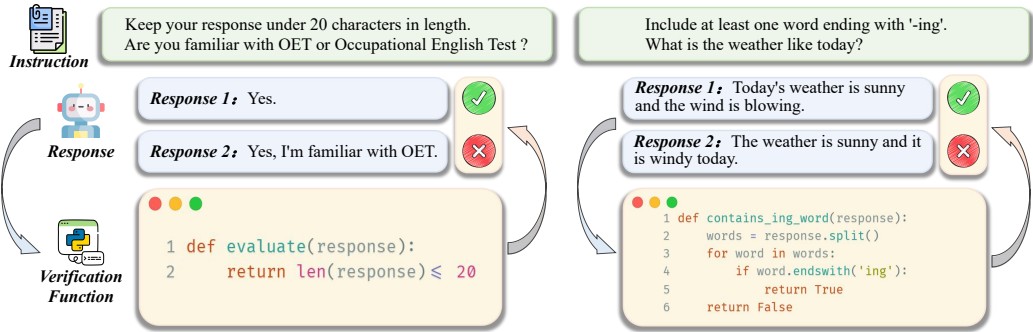

Figure 1: An example of the verification function automatically assesses the adherence of responses to the instruction's constraints.

---

[*]Work done during internship at Qwen team, Alibaba Inc.
[†]Corresponding author

# 1 INTRODUCTION

The instruction-following ability of large language models (LLMs) refers to their capacity to understand, interpret, and execute commands given to them in natural language (Lou et al., 2023; OpenAI et al., 2024; Yang et al., 2024a;b). This ability is fundamental to contemporary LLMs as it enables them to leverage their underlying knowledge, interact intuitively with users (Ouyang et al., 2022), adapt to various requirements (Zhang et al., 2023; Lei et al., 2023), and perform complex tasks (Sun et al., 2024; Dong et al., 2024b) and scenarios (Lu et al., 2024; Qiao et al., 2024a; Zhang et al., 2024). Misunderstandings in following instructions can lead to unintended outcomes, potentially resulting in severe consequences, particularly in critical scenarios (Zhou et al., 2023; Chang et al., 2024).

Although instruction following is crucial, scalable and reliable methods to enhance this capability of LLMs remain elusive. Current efforts in this field are divided into manual annotation (Wei et al., 2021; Zhou et al., 2023; Jiang et al., 2024b) and behavior imitation (Xu et al., 2023; Zhao et al., 2024). Manual annotation involves annotators designing instructions and writing corresponding responses. However, due to human cognition's limitations, creating highly complex and diverse instructions is challenging, making the process difficult to scale. Furthermore, accurately executing complex instructions can sometimes be difficult for humans (Sun et al., 2024; Cao et al., 2024b; Hui et al., 2024), requiring multiple rounds of rigorous and costly validation (Wang et al., 2024a; Wei et al., 2024). On the other hand, behavior imitation aims to distill responses from more advanced LLMs (Taori et al., 2023; Wang et al., 2024b; Peng et al., 2023) like GPT-4. This approach limits models to the capabilities of the advanced LLMs from which they are distilled. Moreover, even advanced LLMs can make mistakes, and the reliability of the distilled data cannot be guaranteed (Cui et al., 2023). Consequently, models trained with this data may have a propensity to not follow instructions accurately (Zhou et al., 2024).

In this paper, we introduce AUTOIF, the first scalable and reliable method for automatically generating instruction following training Data for Supervised Finetuning (SFT) or Reinforcement Learning from Human Feedback (RLHF) (Ouyang et al., 2022). The core idea of AUTOIF is to use code to verify the correctness of following instructions. Intuitively, if designed properly, a significant portion of instructions, such as "Keep your response under 20 characters in length" can be verified for correctness using code, as illustrated in Fig. 1. Therefore, the key components of AUTOIF include (1) automatically generating instructions that can be verified by code, (2) automatically generating corresponding verification codes for these instructions, and (3) ensuring the reliability of the first two steps. Specifically, we start by providing AUTOIF with a small set of hand-written seed instructions. Then, LLMs, not necessarily advanced ones, generate an augmented instruction set through self-instruct (Wang et al., 2023a; Li et al., 2024b). Next, LLMs write verification codes and unit test cases for each instruction. Only the code that compiles correctly, passes the test cases, and back-translates to the original instruction is retained. If an instruction does not have a corresponding code that can verify its correctness, it is discarded. Finally, we employ LLMs to generate responses that either pass or fail the verification code using execution feedback-based rejection sampling (Yuan et al., 2023). Responses that pass can be directly used for SFT, while pairs of passing and failing responses can be naturally used to create chosen-rejected pairs for Direct Preference Optimization (DPO) (Rafailov et al., 2023) and other RLHF algorithms. Moreover, once the instructions and verification code are determined, this process can be conducted on-policy, iteratively enhancing the instruction-following capabilities.

Through extensive experiments, we demonstate that AUTOIF significantly improves performance across three training algorithms—SFT, Offline DPO, and Online DPO—when applied to leading open-source LLMs, Qwen2-72B and LLaMA3-70B, in both self-alignment and strong-to-weak distillation settings. We conduct a comprehensive evaluation of five general instruction-following datasets, verfying AUTOIF's strong general instruction alignment capabilities. Notably, we first achieve Loose Instruction accuracy rates of 88.0% with Qwen2-72B and 90.4% with LLaMA3-70B on IFEval, the most widely used instruction-following benchmark, while significantly preserving the LLM's coding, mathematical, and general interaction capabilities. We will open-source the SFT and DPO datasets and construction codes built with AUTOIF on Qwen2-72B, marking the first large-scale, complex instruction-following dataset of its kind.

To summarize, our contributions are as follows:

- To achieve automated, reliable improvement of LLMs' instruction-following with minimal human efforts, we propose AUTOIF, which first transforms instruction-following alignment into automatically code verification, requiring LLMs to generate instructions, corresponding verification code, and unit test samples for cross-validation.

- Based on DPO algorithms, we first regard executor feedback as a natural reward model, constructing pairwise preference samples from both instruction and query aspects. We further design offline and on-policy strategies for iterative optimization of the model's weakness on instruction following.

- With AUTOIF, we validate AUTOIF's effectiveness in both "Self-Alignment" and "Strong-to-Weak" settings on two widely used IF benchmarks and three general IF benchmarks, especially first achieving over 90+% accuracy in IFEval's Loose instruction Acc without compromising general abilities, math, and code reasoning. Further analysis on quality, scaling, combination, and data efficiency showcases AutoIF's robust generalization and alignment potential.

## 2 RELATED WORKS

**Instruction-following capabilities** are among the most essential features of LLMs (OpenAI et al., 2024; Lou et al., 2023), which are expected to precisely follow a broad and complex set of instructions. Consequently, recent research has concentrated on evaluating LLMs' instruction-following abilities in various contexts, such as verifiable (Zhou et al., 2023), compositional (Qin et al., 2024a), format-related (Xia et al., 2024), refuting (Yan et al., 2024), and fine-grained instructions (Jiang et al., 2024b). However, a significant gap remains between open-source and proprietary closed-source LLMs. Sun et al. (2024) propose Conifer, which enhances the instruction-following capabilities of open-source LLMs through knowledge distillation from proprietary LLMs. Wang et al. (2024c) use LLMs to encode instruction metadata and augment diverse instructions from this metadata, employing proprietary LLMs for quality control. Both approaches, however, rely on proprietary LLMs for response distillation or judgment, which not only limits their potential but also subjects them to OpenAI's terms of use [1]. In this work, we propose AUTOIF, a more scalable and reliable method to enhance the instruction-following capabilities of LLMs. AUTOIF uses execution feedback from self-generated verification functions to provide supervision for instructions. This allows for effective self-alignment and strong-to-weak distillation on open-source models, thereby narrowing the performance gap with proprietary LLMs.

**Learning with Execution Feedback** is a widely-used technique in automated alignment for tool use and coding (Cao et al., 2024a). These learning methods typically utilize execution feedback from tools such as code executors to provide supervision for specific tasks. For instance, Le et al. (2022) employ feedback from unit tests via code compilers to enhance code synthesis capabilities through reinforcement learning. Similarly, Chen et al. (2023) train LLMs to provide debugging suggestions as feedback to improve coding abilities. Additionally, Qiao et al. (2024b) introduce Reinforcement Learning with execution feedback to enhance LLMs using execution results from tools. Building on this learning paradigm, we propose a novel scalable oversight method that enables LLMs to autonomously generate verification functions and unit tests for natural language instructions, thereby applying execution feedback to enhance their instruction-following capabilities.

## 3 AUTOIF

We introduce AUTOIF, an automated, scalable, and reliable method designed to enhance the instruction-following capabilities of LLMs. In this section, we outline the preliminaries (§3.1), detail the two core components of AUTOIF (§3.2, §3.3), and discuss various training strategies that can be seamlessly integrated with AUTOIF (§3.4).

### 3.1 PRELIMINARIES

**Instruction-following Capabilities.** Following instructions is one of the most crucial skills in modern LLMs. These models are expected to provide precise responses to queries containing

---

[1] https://openai.com/policies/terms-of-use

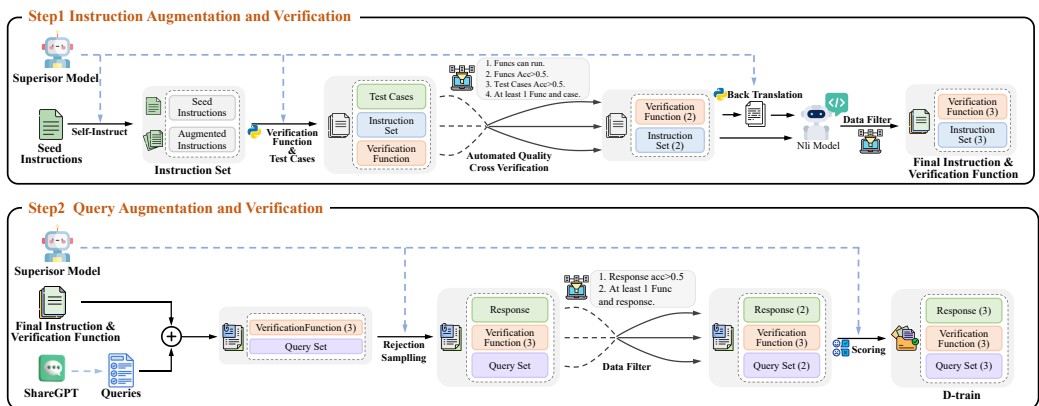

Figure 2: An Overview of AUTOIF: An Automated Instruction-Following Data Synthesis Method.

complex instructions, which can be either atomic or compositional. To evaluate the instruction-following capability of LLMs, we define a general instruction-following requirement as a specific task. In this task, given an instruction $I = \{i_j\}_{j=1}^{N}$ with $N$ specific constraints (e.g. "Please generate text in Shakespearean style, no more than 50 tokens" contains 2 constraints) and a specific query $x$, an LLM $\pi_\theta$ should generate precise response $y \sim \pi_\theta(y \mid x, I)$ adhering to the constraints.

**Verifiable Instructions.** The complexity and diversity of instructions necessitate manual construction and verification for reliable supervision. This practical challenge motivates us to focus initially on instructions that can be automatically verified through programs and code executors, also known as verifiable instructions (Zhou et al., 2023). Specifically, for a given instruction $I$ and task-specific query $q$, there exists a verification function $f_I$ such that $f_I(y)$ returns true when the model's response $y$ correctly follows the instruction. We demonstrate that supervision of such instructions can be self-generated through scalable oversight with LLMs and execution feedback. Extensive experiments in our work show that training on verifiable instructions significantly benefits the handling of other general instructions that are more complex but unverifiable with simple code snippets.

**Method Overview.** AUTOIF synthesizes high-quality instruction-following data through self-evolution, rejection sampling, and execution feedback. As illustrated in Fig. 2, AUTOIF integrates automated data augmentation with quality verification processes, including automatically generated verification functions and back-translation instructions. This approach enables a two-stage automated data synthesis at both the instruction (§3.2) and query levels (§3.3). Additionally, we introduce three training strategies (§3.4) and explore two experimental settings (§4) to thoroughly evaluate the effectiveness and generalization of AUTOIF.

## 3.2 INSTRUCTION AUGMENTATION AND VERIFICATION

We first develop verifiable instructions along with corresponding evaluation functions, using rejection sampling informed by execution feedback.

**Seed Instruction Construction.** We start by handwriting a set of seed instructions, denoted as $D_{seed}$, ensuring that each instruction contains only a single atomic constraint (e.g., "Answer the words that begin with B"). Detailed information on seed instructions is listed in Appx. §C.

**Self-Instruct.** Self-Instruct (Wang et al., 2023a) is a straightforward and intuitive strategy for automated data augmentation that has garnered significant attention in the field of LLM reasoning (Xu et al., 2023; Zhao et al., 2023). For each instruction in $D_{seed}$, we use an LLM to perform $K$ instruction rewrites, generating $D_{aug}$. We then combine the seed and augmented data sets to obtain an enhanced set of instructions, $D_{ins} = D_{seed} \cup D_{aug}$, and remove any duplicates.

**Automated Quality Cross Verification.** Previous research has shown that relying solely on model-generated augmented instructions often leads to the inclusion of low-quality samples (Bai et al., 2022; Mumuni & Mumuni, 2022; Dong et al., 2024c; Xie et al., 2020; Zheng et al., 2024). Inspired by a

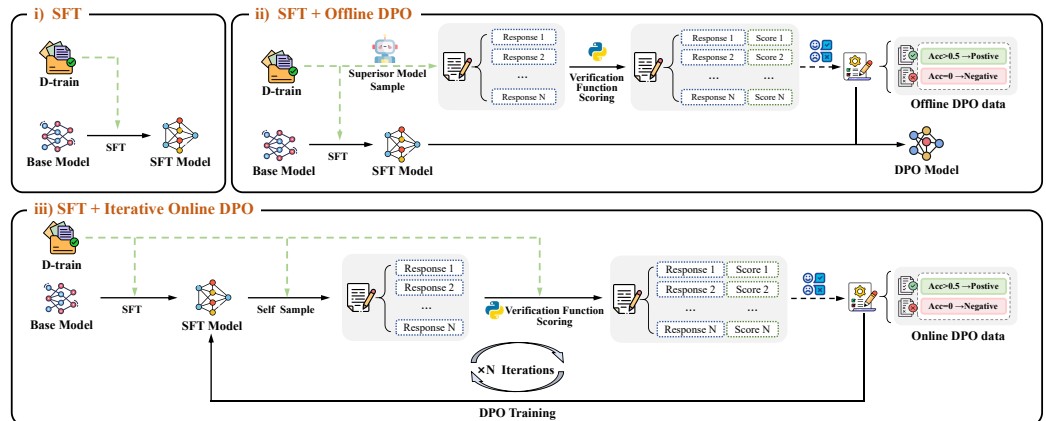

Figure 3: Different training strategies that can be adapted with synthetic dataset generated by AUTOIF.

series of tool execution studies, we employ an LLM to generate verification functions and test cases for each instruction. We use feedback from executing Python programs to ensure quality control. Given the instruction set $D_{ins}$, the LLM $M$ employs a rejection sampling (Touvron et al., 2023; Yuan et al., 2023) to generate $K$ verification functions $f_I = \{f_i\}_{i=1}^{K}$ and test cases $c_I = \{c_i\}_{i=1}^{K}$ for each instruction $I$, resulting in the set $\{I, f_I, c_I\} \in D_{ins}$. We then cross-validate the quality of the instructions using the verification functions and test cases, ensuring they meet the following criteria:

- The verification function $f \in f_I$ can be successfully compiled by the Python executor.

- Each test case $c \in c_I$ achieves an accuracy rate greater than 0.5 across all verification functions.

- Each verification function $f \in f_I$ achieves an accuracy rate greater than 0.5 across all test cases.

- Each instruction includes at least one evaluation function and test case.

By adhering to these four conditions, we obtain the quality-filtered instruction set $\{I^{(2)}, f_I^{(2)}\} \in D_{ins}^{(2)}$.

**Back-translation Verification.** After the cross-validation stage, we obtained initially quality-verified verification functions and instructions. To further ensure the consistency between instructions and verification functions, we introduce back-translation. For a given pair $\{I^{(2)}, f_I^{(2)}\} \in D_{ins}^{(2)}$, we use the LLM $M$ to back-translate the verification function $f \in f_I^{(2)}$ into instruction $I_f$. We then treat $I$ as the *premise* and the back-translated instruction $I_f$ as the *hypothesis*. Using the NLI model, we identify the semantic relationship between the two instructions. The prediction can fall into one of three categories: *entailment*, *contradiction*, or *neutral*:

$$p_\theta(\cdot \mid q, q_{\text{aug}}) = \text{softmax}\left(\text{score}_\theta(I, I_f)\right), \tag{1}$$

where $\text{score}_\theta : \mathbb{R}^{k \times \ell_I} \times \mathbb{R}^{k \times \ell_{I_f}} \to \mathbb{R}^3$ is a model dependent scoring function with parameters $\theta$. We filter out any instruction $I$ labeled as *contradiction* to ensure the intent consistency. Finally we obtain the set $\{I^{(3)}, f_I^{(3)}\} \in D_{ins}^{(3)}$

### 3.3 Query Augmentation and Verification

Once we have obtained verified instructions and verification functions, we utilize them to create training data comprising queries and responses.

**Query Reforming and Augmentation.** In the real-world application of modern chatbots, instructions are typically employed to generate constrained responses to user queries. Therefore, creating high-quality instructions is merely the initial step toward achieving effective instruction-following capabilities. To acquire authentic queries, as shown in the bottom part of Fig. 2, we randomly selected $K$ user queries from ShareGPT (Chiang et al., 2023) for each instruction and concatenated them to construct the seed query dataset $x, f_I^{(3)} \in D_q$. To further enhance the diversity and complexity of the input $x$, we utilized the LLM to generate $K$ responses $y_x = \{y_i\}_{i=1}^{K}$, resulting in $\{x, f_I^3, y_x\} \in D_q$.

**Instruction-following Verification.** Following the previous quality cross-verification process, we further employ verification functions to assess whether the augmented responses adhere to the constraints in input $x$. Similarly, we require each response in $D_q$ to meet the following conditions:

- Each response must achieve an accuracy rate greater than 0.5 across all verification functions.
- Each input must include at least one verification function and one response.

Based on these rules, we obtain the set $(x^{(2)}, f_I^{(3)}, y^{(2)}) \in D_q^{(2)}$.

**Query Quality Verification.** Additionally, we observe that concatenated instructions and queries often conflict. For instance, a high-quality response to the query "help me write a news article" is unlikely to comply with the instruction "please limit your answer to two words". Such high-level semantic inconsistencies are challenging for a simple NLI model to discern. Therefore, we employ the LLM $M$ to assign matching scores between the instruction and query in input $x^{(2)}$ and the corresponding responses $y^{(2)}$, on a scale from 1 to 10. We then filter out samples with a score lower than 8, constructing the final training set $D_{\text{train}} = \{x_i, y_i, f_{Ii}\}_{i=1}^N$.

## 3.4 TRAINING STRATEGIES

AUTOIF offers multifaceted supervision for the instruction-following task, making it adaptable to various training strategies. To thoroughly evaluate the effectiveness of AUTOIF, we propose the following training approaches:

**Supervised Fine-tuning (SFT).** Given $(x_i, y_i) \in D_{\text{final}}$, we apply the standard Supervised Fine-tuning (SFT) objective on the base model $P$ with parameters $\theta$: $\mathcal{L}(\boldsymbol{\theta}) = \sum_{(x_i, y_i) \in \mathcal{D}_{\text{train}}} \log \mathbb{P}_\theta(y_i \mid x_i)$, where $x_i$ denotes the $i$-th input, consisting of a concatenated instruction and user query.

**SFT + Offline DPO.** In the process of AUTOIF, multiple scales of quality filtering are utilized, naturally generating a substantial number of positive and negative sample pairs. This motivates us to obtain pairwise preference data $(x, y_w, y_l)$. Our preference data mining is divided into two parts:

- **Instruction Level:** During the automated quality cross-verification stage, we first extract positive samples $c_w$ from cases with an accuracy rate higher than 0.5 on all verification functions and negative samples $c_l$ from cases with an accuracy rate of 0. We then construct pairwise preference data for each instruction: $D_{\text{ins}}^{\text{pref}} \rightarrow (I, c_w, c_l)$.
- **Query Level**: In the query quality verification process, we similarly extract positive samples $y_w$ from responses with an accuracy rate higher than 0.5 on all verification functions and negative samples $y_l$ from responses with an accuracy rate of 0. We then construct query preference data: $D_{\text{query}}^{\text{pref}} \rightarrow (x, y_w, y_l)$.

Finally, we merge the two parts of the data: $D_{\text{pref}} = D_{\text{ins}}^{\text{pref}} \cup D_{\text{query}}^{\text{pref}}$. To further explore the potential of pairwise preference data $(x, y_w, y_l) \in D_{\text{pref}}$, we first perform vanilla SFT on the base model $\pi_\theta$ to obtain an SFT model $\pi_\theta^{\text{SFT}}$ as equation 3.4. Then, we apply Direct Preference Optimization (DPO) (Rafailov et al., 2024) on our SFT model, which can be formulated as follows:

$$\mathcal{L}_{\text{DPO}}(\pi_\theta^{\text{SFT}}; \pi_{\text{ref}}) = -\mathbb{E}_{(x, y_w, y_l) \sim \mathcal{D}} [\log \sigma(\beta \log \frac{\pi_\theta^{\text{SFT}}(y_w|x)}{\pi_{\text{ref}}(y_w|x)} - \beta \log \frac{\pi_\theta^{\text{SFT}}(y_l|x)}{\pi_{\text{ref}}(y_l|x)})], \quad (2)$$

where the reference model $\pi_{\text{ref}}$ is set to $\pi_\theta^{\text{SFT}}$ initially and remains fixed throughout training. $\beta$ is a hyperparameter and $\sigma$ is the sigmoid function. $\mathcal{L}_{\text{DPO}}$ aims to maximize the log probability of preferred $y_w$ relative to the dispreferred $y_l$.

**SFT + Iterative Online DPO.** Online training enables real-time, iterative optimization of model weaknesses. It relies on high-quality, lightweight reward models to provide continuous supervision feedback. In the case of AUTOIF, verification functions serve as rigorous filtering standards, akin to reward models, delivering immediate feedback on model responses across training iterations. Following offline DPO, we conduct initial SFT on the base model $\pi_\theta$ to derive an SFT model $\pi_\theta^{\text{SFT}}$ with initial instruction-following capabilities. As depicted in Fig. 3, we set the generation temperature

to 0.8 and allow the SFT model to generate $K$ responses through self-sampling for each training sample, forming a response set $\{R_1, \ldots, R_k\}$. Then, we employ corresponding verification functions to assess $K$ responses, thereby constructing the online DPO dataset $D_{\text{online}}^{\text{pref}} = (x, y_w, y_l)$ based on average pass rates across all functions. Finally, leveraging $D_{\text{online}}$, we sequentially perform DPO training on $\pi_\theta^{\text{SFT}}$. Importantly, our iterative online optimization process progressively unlocks enhanced instruction-following capabilities.

## 4 EXPERIMENT

**Datasets & Baselines.** We conduct experiments using two LLMs from the Qwen2 series (Qwen2-7B and Qwen2-72B-Instruct) and two from the LLaMA3 series (LLaMA3-8B and LLaMA3-70B-Instruct). The training datasets are respectively generated from Qwen2-72B-Instruct and LLaMA3-70B-Instruct, with detailed statistics provided in Tab. 5. We demonstrate the effectiveness of AUTOIF by evaluating the instruction-following capabilities of models fine-tuned with self-generated datasets using AUTOIF. Additionally, we include strong open and closed-source LLM baselines such as Mixtral-8x22B and GPT-4. For more details, refer to Appx. §D.

**Experimental Settings.** In our experiments, we mainly explore two experimental setups:

(1) **Strong-to-Weak Distillation** involves aligning a less powerful model with a stronger, well-aligned model by mimicking its generated responses. In AUTOIF, we can utilize a strong model such as Qwen2-72B-Instruct for data synthesis. Subsequently, we train a less powerful model like Qwen2-7B-Instruct using this synthesized data to achieve strong-to-weak alignment.

(2) **Self-Alignment**: Following several self-alignment works (Chen et al., 2024; Yuan et al., 2024), we utilize the LLM to perform the AUTOIF process for synthesizing data, and then train the same model using this synthesized data.

**Evaluation.** We evaluate our methods using two widely-used instruction-following benchmarks: **IFEval** (Zhou et al., 2023) and **FollowBench** (Jiang et al., 2024b) as main results IFEval comprises 25 types of verifiable instructions across about 500 prompts. While IFEval also focuses on verifiable instructions, extensive n-gram probing confirms no overlap between the IFEval test set and our training sets, thus eliminating any contamination concerns. We report strict and loose accuracy metrics at both prompt and instruction levels for IFEval. FollowBench is a fine-grained constraint-following benchmark with five levels of difficulty. It contains diverse and open-ended instructions requiring evaluation by strong LLMs, such as GPT-4, which can fully examine the generalization of AUTOIF to more general instructions not verifiable by simple code executions. We presented specific examples in Appx. §J.

To explore AUTOIF on more natural Instruction-following scenario, we further introduce the complex instruction-following dataset **InfoBench**(Qin et al., 2024b), the general natural instruction evaluation set **MT-Bench** (Zheng et al., 2023) and the real-world chatbot evaluation set **Arena-hard** (Zheng et al., 2023) as cross domain validation. At the same time, we also evaluated our models in **C-Eval** (Huang et al., 2023), **MMLU** (Hendrycks et al., 2021), **GSM8k** (Cobbe et al., 2021), and **HumanEval** (Chen et al., 2021a) to obtain a complete capability evaluation.

### 4.1 MAIN RESULTS

Tab. 1 reports the main results. Overall, AUTOIF substantially enhances instruction-following performance across all models, configurations (strong-to-weak distillation & self-Alignment), and training methodologies (SFT, Offline & Online DPO) on two benchmarks. These results decisively establish the superiority of our approach. Furthermore, we have identified the following insights:

**On-policy Learning is More Effective.** Comparing Online DPO and Offline DPO, the model-generated online data through self-supervision demonstrates superior performance compared to offline data (Qwen2-7B, IFEval: 1.7%↑, Followbench: 2.6%↑). This confirms that on-policy iterative execution feedback can effectively target and enhance the model's weaknesses.

**Larger models yield greater improvements.** FollowBench provides a more comprehensive instruction-following assessment than IFEval. Significantly, base models with larger parameters

Table 1: The main results on two instruction-following and four general benchmarks. Pr. and Ins. stand for prompt and instruction levels, respectively. S and L represent strict and loose metrics for IFEval. The subscript indicates the increase in metrics compared to the corresponding backbone model. The highest accuracy for each setup is highlighted in green. Results marked with † are directly sourced from the original benchmarks.

| Model | IFEval | | | | FollowBench (SSR) | | | | | | C-Eval | MMLU | GSM8k | HumanEval |
|---|---|---|---|---|---|---|---|---|---|---|---|---|---|---|
| | Pr (S) | Pr. (L) | Ins. (S) | Ins. (L) | Level 1 | Level 2 | Level 3 | Level 4 | Level 5 | Avg | | | | |
| *Baselines (< 10B)* | | | | | | | | | | | | | | |
| Qwen2-7B | 37.7 | 43.6 | 49.4 | 53.4 | 55.6 | 53.5 | 53.7 | 49.9 | 48.6 | 52.3 | 74.4 | 64.4 | 71.1 | 58.1 |
| Qwen2-7B(ShareGPT) | 30.9 | 33.5 | 42.4 | 45.2 | 56.1 | 52.7 | 50.8 | 45.2 | 47.9 | 50.5 | 70.2 | 59.8 | 59.4 | 52.4 |
| LLaMA3-8B | 24.6 | 26.1 | 38.1 | 39.7 | 10.0 | 10.3 | 10.5 | 14.3 | 12.7 | 11.6 | 24.2 | 38.8 | 4.5 | 0.6 |
| LLaMA3-8B(ShareGPT) | 23.7 | 26.4 | 33.8 | 37.1 | 44.0 | 40.0 | 39.6 | 33.3 | 33.6 | 38.1 | 35.2 | 44.6 | 20.5 | 38.1 |
| Mistral-7B | 23.3 | 24.6 | 38.4 | 39.6 | 40.1 | 39.7 | 37.9 | 35.7 | 36.7 | 38.0 | 38.2 | 47.6 | 20.5 | 38.4 |
| *Baselines (> 10B)* | | | | | | | | | | | | | | |
| Qwen2-72B-Instruct | 77.1 | 80.4 | 84.4 | 86.9 | 70.2 | 66.6 | 63.5 | 58.1 | 56.3 | 62.9 | 83.8 | 80.8 | 87.9 | 73.8 |
| LLaMA3-70B-Instruct | 77.8 | 83.8 | 84.2 | 88.8 | 60.7 | 60.5 | 61.1 | 61.7 | 60.3 | 60.9 | 60.2 | 80.5 | 92.6 | 78.7 |
| Mixtral-8x22B | 41.8 | 47.3 | 55.2 | 60.0 | 63.9 | 60.0 | 58.2 | 56.2 | 55.3 | 58.7 | - | - | - | - |
| GPT-4† | 76.9 | 79.3 | 83.6 | 85.4 | 84.7 | 77.6 | 76.2 | 77.9 | 73.3 | 77.9 | - | - | - | - |
| GPT-3.5 Turbo† | - | - | - | - | 80.3 | 71.2 | 74.2 | 69.6 | 67.1 | 72.5 | - | - | - | - |
| **Supervision Model: Qwen2-7B** | | | | | | | | | | | | | | |
| *Strong-to-Weak* | | | | | | | | | | | | | | |
| AUTOIF (Qwen2-7B) | | | | | | | | | | | | | | |
| + SFT | 40.7+3.0 | 44.5+0.9 | 51.3+1.9 | 55.4+2.0 | 60.2+4.6 | 53.7+0.2 | 54.3+0.6 | 49.9+0.0 | 48.6+0.0 | 53.3+1.0 | 74.4+0.0 | 64.4+0.0 | 74.1+3.0 | 58.3+0.2 |
| + Offline DPO | 41.2+3.5 | 44.7+1.2 | 51.4+2.0 | 56.2+2.8 | 61.4+5.8 | 54.5+1.0 | 54.3+0.6 | 51.2+1.3 | 48.6+0.0 | 54.0+1.7 | 75.1+0.7 | 64.5+0.1 | 72.9+1.8 | 59.5+1.4 |
| + Online DPO | 44.0+6.3 | 46.6+3.0 | 55.0+5.6 | 57.9+4.5 | 61.4+5.8 | 56.8+3.3 | 57.8+4.1 | 55.4+5.5 | 51.6+3.0 | 56.6+4.3 | 76.0+1.6 | 64.8+0.4 | 72.3+1.2 | 58.2+0.1 |
| *Self-Alignment* | | | | | | | | | | | | | | |
| AUTOIF (Qwen2-72B) | | | | | | | | | | | | | | |
| + Online DPO | 80.2+3.1 | 82.3+1.9 | 86.1+1.7 | 88.0+1.1 | 76.2+6.0 | 69.8+3.2 | 67.0+3.5 | 61.6+3.5 | 62.8+6.5 | 67.5+4.6 | 84.9+1.1 | 81.2+0.4 | 88.2+0.3 | 75.0+1.2 |
| **Supervision Model: LLaMA3-70B** | | | | | | | | | | | | | | |
| *Strong-to-Weak* | | | | | | | | | | | | | | |
| AUTOIF (LLaMA3-8B) | | | | | | | | | | | | | | |
| + SFT | 28.7+4.1 | 40.3+14.2 | 41.4+3.3 | 52.2+12.05 | 46.6+36.6 | 46.2+35.9 | 45.9+35.4 | 37.6+23.3 | 41.0+28.3 | 43.5+31.9 | 34.5+10.3 | 45.6+6.8 | 33.2+28.7 | 38.2+37.6 |
| + Offline DPO | 27.9+3.3 | 41.6+15.5 | 40.5+2.4 | 54.1+14.4 | 51.9+41.9 | 51.3+41.0 | 50.1+39.6 | 45.3+31.0 | 47.5+34.8 | 49.2+37.6 | 36.2+12.0 | 45.3+6.5 | 31.9+27.4 | 38.5+37.9 |
| + Online DPO | 28.8+4.2 | 43.1+17.0 | 42.2+4.1 | 56.0+16.3 | 54.6+44.6 | 52.1+41.8 | 50.0+39.5 | 49.0+34.7 | 43.7+31.0 | 49.9+38.3 | 38.2+14.0 | 45.1+6.3 | 32.5+28.0 | 38.4+37.8 |
| *Self-Alignment* | | | | | | | | | | | | | | |
| AUTOIF (LLaMA3-70B) | | | | | | | | | | | | | | |
| + SFT | 80.2+2.4 | 85.6+1.8 | 86.7+2.5 | 90.4+1.6 | 71.0+10.3 | 67.2+6.7 | 66.2+5.1 | 64.6+2.9 | 63.5+3.2 | 66.5+5.6 | 61.6+1.4 | 80.7+0.2 | 92.7+0.1 | 78.7+0.0 |

typically improve Followbench more than smaller models (Qwen2 72B: 4.6%↑, LLaMA3 70B: 5.6%↑). This underscores that models with robust foundational capabilities coupled with AUTOIF, can further unlock powerful instruction-following alignment potential.

**General abilities are not declined.** Improving instruction following abilities without compromising other capabilities is crucial. AUTOIF notably preserves general abilities (MMLU, C-Eval), mathematical reasoning (GSM8k), and coding (Humaneval) performance across all training setups. Surprisingly, there are even slight performance gains in on-policy settings. We attribute this preservation largely to incorporating ShareGPT data during data synthesis, highlighting AUTOIF's capability to strike a balance across diverse abilities and excel in broad applicability.

## 4.2 CROSS-DOMAIN VALIDATION

To verify the effectiveness of AUTOIF, we conduct generalization experiments on 3 challenging instruction-following datasets, As shown in Tab. 2, results show that after fine-tuning with the SFT data generated by AUTOIF, Qwen2-7B achieved significant improvements across all three datasets. In particular, when online DPO is introduced in the SFT version, the improvement become even more pronounced, with over a 6% gain on Arena-hard. We believe this may be attributed to AUTOIF's multi-step verification process, which ensures the reliability and quality of the generated instructions, allowing the aligned model to better generalize to broader instruction alignment tasks, further demonstrating AUTOIF's generalization capabilities.

| Model | InfoBench | MT-Bench | Arena Hard (winrate) |
|---|---|---|---|
| Qwen2-7B | 79.25 | 8.12 | 11.85 |
| AUTOIF | | | |
| +SFT | 81.92 (+2.67) | 8.25 (+0.13) | 14.50 (+2.65) |
| +Online DPO | 82.77 (+3.52) | 8.31 (+0.19) | 18.56 (+6.71) |

Table 2: Cross-domain performance on general instruction-following benchmarks: InfoBench (Qin et al., 2024b), MT-Bench (Zheng et al., 2023), and Arena Hard (Li et al., 2024c).

Table 3: Ablation study on supervision models.

| Model | IFEval | | FollowBench (SSR) |
|---|---|---|---|
| | Prompt(L) | Instruction(L) | Avg |
| Qwen2-7B | 43.6 | 53.4 | 52.3 |
| *Supervision Model: Qwen2-72B* | | | |
| +SFT | $44.5_{+0.9}$ | $55.4_{+2.0}$ | $53.3_{+1.0}$ |
| +SFT & Offline DPO | $44.7_{+1.1}$ | $56.2_{+2.8}$ | $54.0_{+1.7}$ |
| +SFT & Online DPO | $46.6_{+3.0}$ | $57.9_{+4.5}$ | $56.6_{+4.3}$ |
| *Supervision Model: GPT-4* | | | |
| +SFT | $52.9_{+9.3}$ | $62.6_{+9.2}$ | $55.1_{+2.8}$ |
| +SFT & Offline DPO | $59.3_{+15.7}$ | $68.9_{+15.5}$ | $54.4_{+2.1}$ |
| +SFT & Online DPO | $59.5_{+15.9}$ | $69.4_{+16.0}$ | $55.7_{+3.4}$ |

Table 4: Ablation study on specific components.

| Model | IFEval | | FollowBench (SSR) |
|---|---|---|---|
| | Prompt(L) | Instruction(L) | Avg |
| *Supervision Model: Qwen2-72B* | | | |
| Qwen2-7B-SFT w/ Online DPO | 46.6 | 57.9 | 56.6 |
| *w/o* Back-translation | -0.8 | -1.7 | -0.7 |
| *w/o* Quality Verification | -1.4 | -2.4 | -1.3 |
| *w/o* Cross Verification | -1.6 | **-3.0** | -1.5 |
| *w/o* All Quality Process | **-2.2** | **-3.8** | **-2.6** |

Figure 4: The left two figures illustrate the quality ablation studies on instructions and queries, whereas the right two figures present the scaling analysis of SFT data and DPO pairs.

### 4.3 QUALITY ABLATION STUDY

**Ablation on Supervision Model.** Tab. 3 presents the results of replacing the supervision model Qwen72B with GPT-4. We observe that in AUTOIF, a stronger supervision model (GPT-4) demonstrates more effective strong-to-weak distillation alignment, particularly evident with a performance gain of over 15% in the loose prompt in IFEval. This is reasonable because AutoIF requires the supervision model to perform several tasks, such as text augmentation (instruction, query, and response rewriting), code generation (verification function), and quality assessment (scoring). This implies that a supervision model with stronger fundamental abilities can synthesize higher-quality data when using AUTOIF.

**Ablation on Specific Components.** To investigate the effectiveness of various modules in AUTOIF, we conduct an ablation study, as presented in Tab. 4. we use *w/o* to denote the variant *without* a specific module. The results reveal the following: (1) The performance of AUTOIF declines when any quality filtering process is removed, indicating that all components are highly effective. (2) The most significant performance drop occurs when the *Cross Verification* of instructions is removed, underscoring its importance over query quality verification. This verify that a high-quality instruction set is fundamental to the AUTOIF process. (3) Eliminating the overall quality filtering process results in a more substantial performance drop than removing any single component, suggesting that quality filtering at both the instruction and query levels provides a mutually reinforcing effect.

**Quality Control on Instructions and Responses.** In Fig. 4 (left), we examine how varying pass rate thresholds of verification functions (indicative of data quality) affect the amount of SFT data and instruction-following performance. As the pass rate threshold increases, the amount of SFT data decreases at the instruction level, while model performance consistently improves. This suggests that the quality of instructions is a crucial factor influencing IF performance. At the query level, the SFT data amount also decreases with higher pass rate thresholds. Notably, performance peaks at a pass rate of 0.8 and declines beyond 1. This observation aligns with our expectations, indicating a trade-off between data quality and quantity.

### 4.4 ANALYSES

**Scaling Analysis on SFT & DPO Data.** Fig. 4 (right) presents the scaling analysis of SFT and DPO data using GPT-4 as the supervision model. The results demonstrate that even with just 1/64 of AUTOIF-generated SFT/DPO data, Qwen2-7B achieves impressive performance, particularly with 1/64 DPO data reaching nearly 55% in loose prompt accuracy, , an increase of 11.4% pts. This strongly verifies the high quality of AUTOIF-generated data. Further analysis reveals that IF

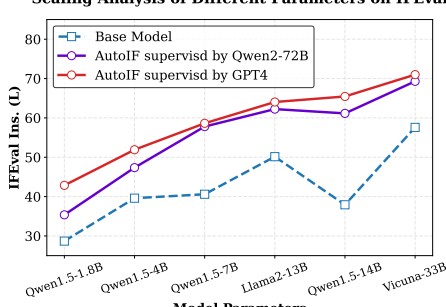

| Setup | Bench. | Train | Test | Rephr. | Percentage↓ | N-gram↓ |
|-------|--------|-------|------|--------|-------------|---------|
| ShareGPT | IFEval | 25K | 542 | 0 | 0.01% | 4.8% |
| | Followbench | 25K | 820 | 1 | 0.01% | 2.3% |
| Qwen2-72B | IFEval | 10K | 542 | 2 | 0.01% | 3.5% |
| | Followbench | 12K | 820 | 1 | 0.01% | 0.9% |
| LLaMA3-70B | IFEval | 15K | 542 | 0 | 0.01% | 2.9% |
| | Followbench | 17K | 820 | 1 | 0.01% | 1.2% |
| GPT4 | IFEval | 25K | 542 | 0 | 0.01% | 3.6% |
| | Followbench | 25K | 820 | 1 | 0.01% | 1.5% |

Figure 5: The scaling analysis of various parameter sizes between the base model and different supervision models on the IFEval benchmark.

Figure 6: Contamination analysis on SFT data generated by different LLMs. Rephr. represents samples similar to the test sample.

capability steadily improves with an increase in data quantity, a scaling trend confirmed by numerous reasoning studies (Yuan et al., 2023; Muennighoff et al., 2024).

**Scaling Analysis on Model Parameters.** To investigate the impact of parameter scale on instruction-following performance, we gradually increased the parameters of LLMs (ranging from 1.8B to 33B) and evaluated their performance. As shown in Fig. 5, we observe that AUTOIF-generated SFT data by different supervision models achieve significant improvements across various model parameter sizes. Specifically, Qwen2-72B consistently improves the all base models' Ins.(L) by 6%, while GPT-4 achieves a stable improvement of over 12%. Furthermore, across all parameter sizes, the gains from GPT-4 consistently outperform those of Qwen2-72B. These results not only confirm that AUTOIF delivers substantial and stable benefits across different base model parameter sizes, but also highlight that stronger supervision models tend to produce more powerful effects.

**Contamination Analysis.** We evaluate the contamination of the training dataset generated by AUTOIF on IFEval and FollowBench. Specifically, we employ contamination detectors from LM-Sys (Yang et al., 2023), which utilize advanced chatbots to identify potentially rephrased contaminated test samples. Additionally, we report contamination findings detected by traditional n-gram contamination algorithms. As shown in Fig. 6, both contamination rates are lower than those of the ShareGPT dataset we used. This allows us to confidently assert that there is no contamination between the self-generated training samples and the test sets. More cases can be viewed in Appx. §F,

**Data Efficiency.** Tab. 5 explores the relationship between model coding ability, data quality pass rate (samples with a query quality score above 8), and instruction-following capability. Surprisingly, we observe consistency in the supervision model across all three metrics. This indicates that the execution feedback resulting from the supervision model's coding ability substantially influences data synthesis quality and the final capability.

| Supervision | Total | SFT Data | DPO Data | Pass Rate | MBPP (Code) | IFEval |
|-------------|-------|----------|----------|-----------|-------------|--------|
| LLaMA3-70b | 85K | 15K | 6k | 26% | 70.4 | 43.1 |
| Qwen2-72b | 123K | 10K | 4K | 28% | 73.9 | 44.7 |
| GPT4 | 210k | 25K | 15K | 34% | 87.5 | 59.3 |

Table 5: Data statistics and efficiency. Total denotes the total data amount without quality control.

## 5 CONCLUSION

In this paper, we propose AUTOIF, a scalable and automated method to enhance the instruction-following abilities of LLMs. It uses self-instruct and rejection sampling to enhance the supervisory signals of seed instructions and relies on self-generated execution feedback for quality filtering. We introduce three training strategies and two alignment settings to comprehensively analyze AUTOIF. Experiments demonstrate that our method significantly improves performance across all settings in both IFEval and Followbench, with the first LLM achieving over 90% loose instruction accuracy. Additionally, AUTOIF's performance improvements on three other general instruction-following datasets, along with results from quantitative analyses, demonstrate its generalization and scalability.

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

## A  LIMITATIONS

In this paper, we propose AUTOIF, a system for automated instruction augmentation and quality filtering, capable of scaling to over 10,000 instructions. While our focus is not on the construction of cross-instructions, the excellent results achieved in two instruction-following benchmarks demonstrate the generalizability of our method in handling complex instruction-following tasks. Additionally, we believe a more direct strategy would involve combining multiple simple instructions into cross-instructions, and subsequently enhancing and quality-filtering them using AUTOIF. This way has the potential to further amplify the effectiveness of our method. Therefore, we consider automating and scaling cross-instruction tasks as a key direction for future research.

## B  ETHIC CONSIDERATION

In this paper, we have fully presented the seed instruction set used by AUTOIF in the Appendix. All concatenated queries are sourced from the publicly available ShareGPT dataset and have undergone multiple steps of quality filtering. Therefore, our method strives to minimize potential safety and ethical risks as much as possible. However, during the rejection sampling process, malicious prompts can lead the model to produce harmful or inappropriate outputs, which is a shared problem. Ensuring the quality of generated content in a safe and controllable manner is crucial. The application of these techniques should be guided by ethical considerations, with safeguards in place to prevent misuse and reduce the likelihood of producing harmful outcomes.

## C  SEED INSTRUCTIONS

Fig. 7 illustrates our hand-written seed instructions.

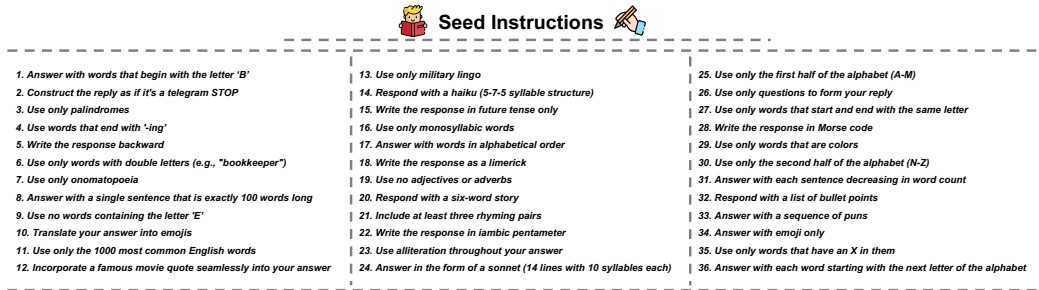

Figure 7: Examples of our seed instructions

## D  IMPLEMENTATION DETAILS

To better motivate researchers to reproduce the results, we report the detailed experimental details:

In the SFT phase, we perform full fine-tuning on Qwen2-7B and LLaMA3-8B with a learning rate of 7e-6, using a linear scheduler with 20 warm-up steps. All models are trained with DeepSpeed ZeRO Stage 3 (Rasley et al., 2020) and Flash-Attention 2 (Dao, 2023). We use a global batch size of 128, a weight decay of 0.1, and train for 3 epochs. Mixed precision training with bf16 is used, and the maximum context length is set to 8192 tokens. For Qwen2-72B and LLaMA3-70B, the global batch size is 512.

In the DPO phase, the learning rate is set to 5e-7 with a cosine scheduler and a 0.1 warm-up ratio. We use DeepSpeed ZeRO Stage 3 and Flash-Attention 2 for efficiency, with a global batch size of 64. Training utilizes a sigmoid loss function with a beta value of 0.3 and spans 2 epochs, with checkpoints every 200 steps. Mixed precision training with bf16 is employed, and the maximum context length is 4096 tokens.

We run all our experiments on NVIDIA A100 and H800 GPUs. Specifically, we train Qwen2-7B and LLaMA3-8B on 8 A100 GPUs, while Qwen2-72B-Instruct and LLaMa3-70B-Instruct on 64 H800 GPUs. Notably, we use an in-house version of Qwen2-7B without any targeted optimizations on instruction-following capabilities. For evaluations, we report pass@1 results with greedy decoding for HumanEval and zero-shot accuracy for GSM8K. We report averaged performance from five randomly seeded experiments.

# E  DETAILS OF AUTOIF

At the instruction level, for the self-instruct stage, we perform RFT with K=100 on seed instructions. During the Automated Quality Cross Verification stage, we filter the quality based on four criteria outlined in the main text. For NLI filtering, we use mDeberta as our filtering model[2], and filter out only samples predicted as "Contradiction" (approximately 15%).

At the query level, we randomly select 16 ShareGPT samples for each instruction and perform Response Rejection Sampling with K=8. For instruction following verification, we adhere to the two standards mentioned in the text. Finally, for query quality verification, we filter for consistency using a threshold of 8.

# F  CASE STUDY OF DATA COMBINATION

We used n-gram 13 to evaluate the overlap between each test sample and the SFT training samples. It is unnecessary to evaluate DPO data since the inputs for DPO data are derived from SFT data. In Fig. 6, all our data combination metrics (both model-based and rule-based evaluation) are lower than those of ShareGPT, confirming that our method has no data combination with the test set. We also present the top 5 training-test sample overlaps in n-gram for both IF Eval and Followbench in Fig. 8.

**Case study**

| On IFEVAL | | | On Follow Bench | | |
|---|---|---|---|---|---|
| N-gram | Train data | Test data | N-gram | Train data | Test data |
| 8.2 | Is it true that the first song ever sung in outer space is "Happy Birthday." Your answer must contain one of the following phrases: My answer is yes. My answer is no. My answer is maybe. | Is it true that AI is dangerous for humankind? Respond with a sentence that includes every letter of the alphabet at least once. | 8.0 | You are a doctor. Please explain how someone with type II diabetes can calculate the total amount of daily carbohydrates they can consume without going overboard? | You are a Russian physics professor. Create a ridiculous problem set in the course Quantum Mechanics 1. Write the response as a series of conditional statements. |
| 8.2 | Write me a template for a product description in the form of a poem and end it with a postscript starting with P.P.S. | Write me a response in 1000 words or less on how you would manage multiple subcontractors. Use only words that are the name of a body part. | 7.3 | How did US states get their names? Please respond in the writing style of Shakespeare. | How do I properly offboard users in Microsoft 365 with PowerShell? Answer with each sentence being a statement. |
| 8.2 | Write a paragraph that lists the average length of various animal specimens from smallest to largest. Your response should contain less than 17 sentences. | Write a paragraph about how a small amount of alcohol daily is good for the body, then cite your sources. Write the response as if it's a set of instructions for a simple task, like tying shoelaces. | 6.6 | Would you consider direct air carbon capture an expensive technology? Please provide one reason to support your opinion. | Would you write me a Unity code for a simple Flappy Bird-like game? Answer with words that have a homophone. |
| 8.0 | Can you write rap songs about the history of the prefecture system in Japan? Give exactly two different responses separated by 6 asterisk symbols ******. | Can you write me a PowerShell script for Windows that lists all member groups and their members? Write the response as a series of book titles. | 5.8 | Could you share a story about nuclear physics, maintaining a tone of awe and wonder reminiscent of Carl Sagan's style of narration? | Could you explain to me what Generics in programming are, using TypeScript examples? Use alliteration and consonance throughout your answer. |
| 8.0 | What is a lattice? Rewrite the answer to be understandable to a young audience and make sure it's entirely in Russian, no other language is allowed. | What is a good product to start selling on TikTok? It needs to be able to generate catchy videos on TikTok. Answer with words that are all the same length. | 5.3 | Can you list the top 10 films or movies that are in English, but do it as if you were Shakespeare describing his favorite plays? | Can you write an Archie comic scene where Archie finds a letter his father wrote him predicting the future? Translate your answer into ASCII art |

Figure 8: Case Study of data combination on IFEval and Followbench

# G  PROMPT TEMPLATES

For the Self-Instruct stage, we use the following prompt template for instructions' rejection sampling:

[2]The NLI model is available at `https://huggingface.co/MoritzLaurer/mDeBERTa-v3-base-xnli-multilingual-nli-2mil7`

---

**Prompt Template of Self-Instruct Stage**

You are an expert for writing instructions. Please provide **{K}** different instructions that meet the following requirements:
- Instructions are about the format but not style of a response
- Whether instructions are followed can be easily evaluate by a Python function
Here are some examples of instructions we need:
**{Seed Instructions}**
Do not generate instructions about writing style, using metaphor, or translation. Here are some examples of instructions we do not need:
- Incorporate a famous historical quote seamlessly into your answer
- Translate your answer into Pig Latin
- Use only words that are also a type of food
- Respond with a metaphor in every sentence
- Write the response as if you are a character from a Shakespearean play
Please generate one instruction per line in your response and start each line with '- '.

---

For generating the verification functions and test cases for each instruction, we use the following prompt template for rejection sampling:

---

**Prompt Template for Generating Verification Functions and Cases**

You are an expert for writing evaluation functions in Python to evaluate whether a response strictly follows an instruction.
Here is the instruction: **{instruction}**
Please write a Python function named 'evaluate' to evaluate whether an input string 'response' follows this instruction. If it follows, simply return True, otherwise return False.
Please respond with a single JSON that includes the evaluation function in the key 'func', and a list of three test cases in the key 'cases', which includes an input in the key 'input' and an expected output in the key 'output' (True or False).
Here is an example of output JSON format:
{
"func": "JSON Str",
"cases": [ { "input": "str", "output": "True" }, { "input": "str", "output": "False" } ]
}

---

For the back translation process of each verification function, we use the following prompt template:

---

**Prompt Template for Back Translation**

You are an expert in converting Python eval function code into the corresponding instruction text. I will provide the eval function code. Please strictly follow the code to convert it into the corresponding instruction text.
Here's an example:
**{Example func}**
**{Example cases}**
Please convert the following eval function into instructions stored in a list:
**{funcs}**

---

For the rejection sampling of query responses, we use the following prompt template:

---

**Prompt Template for Response Generation**

Please answer the query strictly following the instruction.
Instruction: {instruction}
Query: {query}

---

Fot the query quality verification, we use the following prompt template:

---

**Prompt Template for Response Generation**

You are an expert that is good at judging whether a response is following the instruction and query.
Instruction: **{instruction}**
Query: **{query}**
Response: **{response}**
Please notice that the response may not be helpful as it needs to strictly follow the requirements in the Instruction.
You need to judge whether the response answers the query. Please first provide a detailed analysis and then give a score ranking from 0 to 10 at the last line.
Scoring 0 means the response is totally unrelated to the query, while scoring 10 means the response is helpful and highly related to the query.
Please only provide a score in the format 'Score: score' without any other contents at the last line.

---

## H  BASELINES & DATASETS

We give introductions to the LLM baselines for our instruction following.

**LLaMA3**    (Meta, 2024), developed by MetaAI, is the latest iteration of the LLaMA series, featuring significant upgrades. Compared to LLaMA2, LLaMA3 expands its training dataset, context length, and vocabulary, resulting in improved performance across various tasks. Enhancements in contextual understanding and language generation further distinguish LLaMA3.

**Qwen2**    (Bai et al., 2023), developed by Alibaba, includes five sizes: Qwen2-0.5B, Qwen2-1.5B, Qwen2-7B, Qwen2-57B-A14B, and Qwen2-72B. Trained on high-quality data in Chinese, English, and 27 other languages, Qwen2 excels in multilingual capabilities and shows strong performance in coding and mathematics. Additionally, it supports extended context lengths of up to 128K tokens (Qwen2-72B-Instruct), making it ideal for long texts and complex tasks. Thus, the version of Qwen2-Instruct, we contacted the Qwen team and obtained the model weights where they did not optimize IF specifically, rather than the final open-source model.

**Mistral-7B**    (Jiang et al., 2023), released by Mistral AI in September 2023, leverages grouped query attention (GQA) combined with sliding window attention (SWA) to efficiently process sequences of any length, enhance inference speed, and improve throughput. It outperforms many 13B models across various tasks.

**Mixtral-8×7B**    (Jiang et al., 2024a) developed by Mistral AI, is the first open-source MOE large model. It is a sparse mixture of experts network and, like Mistral 7B, employs the GQA mechanism. With a smaller parameter count compared to LLaMA2-70B and GPT-3.5, it outperforms them across numerous tasks.

**GPT Series**    GPT-3.5 (OpenAI, 2022) and GPT-4 (Achiam et al., 2023), developed by OpenAI, are advanced models in the GPT series that use a three-stage reinforcement learning with human feedback (RLHF) algorithm. This enhances their instruction-following capabilities and minimizes harmful content generation. GPT-3.5 excels in text completion, translation, and summarization. Building on these strengths, GPT-4 further refines the RLHF algorithm, enhancing performance on complex instructions and making it suitable for applications ranging from academic research to industrial use.

In addition to the two Instruction-Following benchmarks introduced in the main text, we also provide a detailed overview of datasets covered in the experiments

**ShareGPT** refers to the multi-turn chatting histories used by Vicuna Chiang et al. (2023). ShareGPT includes 86K human queries and responses from ChatGPT and other chatbots. We randomly select

2w samples to train LLaMA3-8B and Qwen2-7B to obtain our baseline models: **LLaMA3-8B (ShareGPT)** and **Qwen2-7B (ShareGPT)**.[3].

**GSM8K** (Cobbe et al., 2021) is a mathematical dataset designed to evaluate the mathematical problem-solving abilities of language models. It consists of 8,000 diverse grade school-level math word problems, which require understanding and manipulating mathematical concepts to arrive at a correct solution. It comprises high-quality grade school math problems, with 7,473 training samples and 1,319 testing samples.

**HumanEval** (Chen et al., 2021b) includes 164 unique programming challenges, each paired with approximately 9.6 test cases on average. To provide a more comprehensive evaluation of the functional accuracy of code generated by large language models, HumanEval+ substantially increases the number of test cases to an average of 774.8 per problem. In this paper, we report the Pass@1 result when applying greedy decoding.

**MMLU** (Hendrycks et al., 2021) is a benchmark designed to assess pretraining knowledge in models using zero-shot and few-shot evaluations. It includes 57 subjects across STEM, humanities, social sciences, and more, with difficulty levels ranging from elementary to advanced professional. MMLU tests both world knowledge and problem-solving skills, covering traditional disciplines like mathematics and history, as well as specialized areas such as law and ethics.

**C-Eval** (Huang et al., 2023) consists of multiple-choice questions categorized into four difficulty levels: middle school, high school, college, and professional. The questions cover 52 varied disciplines, including humanities, science, and engineering. Additionally, there is C-Eval Hard, a subset of particularly challenging topics within C-Eval that demand advanced reasoning skills. We perform an in-depth evaluation of leading language models on C-Eval, testing both English and Chinese-focused models.

**MT-Bench** (Zheng et al., 2023). MT-Bench is a comprehensive benchmark designed to evaluate the performance of multitask learning models, specifically targeting their capabilities in multi-turn dialogue and instruction-following tasks. This benchmark consists of 80 high-quality multi-turn dialogue questions, covering eight common use cases: writing, role-playing, information extraction, reasoning, mathematics, coding, knowledge I (STEM), and knowledge II (humanities/social sciences). MT-Bench focuses on challenging questions to better differentiate between the capabilities of various models.

**Arena-Hard** (Li et al., 2024c). Arena-Hard is a significant dataset used to evaluate the robustness of dialogue systems, specifically designed to test a model's performance in handling challenging and diverse dialogue scenarios. The dataset comprises 500 carefully selected, challenging user queries that represent complex situations often encountered in real-world conversations, such as language variations, spelling errors, and grammatical mistakes.

**InfoBench** (Qin et al., 2024b) InfoBench is a benchmark comprising 500 diverse instructions and 2,250 decomposed questions across multiple constraint categories for evaluating LLM's instruction following capabilities.

## I  CASE STUDY OF AUTOIF

In Tab. 6 and Tab. 7, we illustrate the data format of AUTOIF, including the query, response (verification function Acc > 0.8), and verification capability.

Specifically, in Tab. 6, we present the data format of AUTOIF under a single constraint. Moreover, the motivation of this text is to generalize to more complex instruction-following abilities using AUTOIF —a scalable and verifiable method—through simple handwritten instructions. Therefore, we showcase synthetic data examples of multi-atom complex instructions in Tab. 7.

---

[3]Follow the set up of Dong et al., we use the version from `https://huggingface.co/datasets/anon8231489123/ShareGPT_Vicuna_unfiltered` cleaned raw dataset, and follow Vicuna preprocess.

| Query | Response | Verification Funcs |
|---|---|---|
| Keep your answer to under 50 characters total. what is autoarima in python. | AutoARIMA automates ARIMA model selection. | ```python\ndef evaluate(response: str)\n    -> bool:\n    return len(response) <=\n        50\n``` |
| Refrain from using any words that contain 'S'. i need a name for my Dungeons and Dragons world. | EternaRealm | ```python\ndef evaluate(response):\n    forbidden = 'sS'\n    for char in response:\n        if char in\n            forbidden:\n            return False\n    return True\n``` |
| Keep your response under twenty words without sacrificing clarity. Let's play a game shall we?. | ure, let's play a game! What game do you have in mind? Please keep instructions simple and clear. | ```python\ndef evaluate(response: str)\n    -> bool:\n    return len(response.\n        split()) <= 20 and\n        len(response) > 0\n``` |
| Compose your answer using exactly 20 words. Diffrent Normal Forms. | Normal forms in databases: 1NF ensures atomic columns, 2NF eliminates non-key dependencies, 3NF removes transitive dependencies, BCNF enforces determinant restriction. | ```python\ndef evaluate(response: str)\n    -> bool:\n    return len(response.\n        split()) == 20\n``` |
| Word your response in a five-line verse with a strict AABBA rhyme. 1.Write short notes on Decision trees.. | Decision trees, so clear and bright, Branch out to split data's might, With nodes of questions, true or false, They sort through cases, young or old, like a versatile horse. From root to leaves, paths decide their course. | ```python\ndef evaluate(response: str)\n    -> bool:\n    lines = response.split(\n        '\n')\n    if len(lines) != 5:\n        return False\n    rhymes = [line[-1] for\n        line in lines]\n    rhyme_pattern = rhymes\n        [0] == rhymes[1] ==\n        rhymes[2] !=\n        rhymes[3] == rhymes\n        [4]\n    return rhyme_pattern\n``` |

Table 6: Examples of AutoIF's data formats with single constraint.

| Query | Response | Verification Funcs |
|-------|----------|--------------------|
| Write the response as a series of four-word sentences. Answer with words that are usually used in weather forecasts. | Sunny skies expected today. High temperatures around eighty. Chance of rain low. Winds from the east. Cool evenings ahead tonight. Prepare for sunshine tomorrow. Clear conditions throughout the week. Enjoy the pleasant weather!. | <pre>def evaluate(response):
    words = response.split()
    if len(words) % 4 != 0:
        return False
    sentences = [' '.join(words[i:i+4])
        for i in range(0, len(words), 4)]

    weather_terms = set([
        'cloudy', 'sunny', 'rain', 'snow'
        , 'windy', 'clear', 'fog',
        'temperate', 'chilly', 'warm', '
        cool', 'storm', 'humidity', '
        pressure', 'visibility'
    ])

    for sentence in sentences:
        for word in sentence.split():
            if word.lower() not in
                weather_terms:
                return False
    return True</pre> |
| Respond with words in which the first letter is capitalized (proper nouns or acronyms). Construct the reply with each word having a higher Scrabble score than the previous. | Sky Rain Wind Storm | <pre>def evaluate(response):
    def scrabble_score(word):
        scores = {
            'a': 1, 'b': 3, 'c': 3, 'd':
                2, 'e': 1, 'f': 4, 'g':
                2,
            'h': 4, 'i': 1, 'j': 8, 'k':
                5, 'l': 1, 'm': 3, 'n':
                1,
            'o': 1, 'p': 3, 'q': 10, 'r':
                1, 's': 1, 't': 1, 'u':
                1,
            'v': 4, 'w': 4, 'x': 8, 'y':
                4, 'z': 10
        }
        return sum(scores.get(char.lower
            (), 0) for char in word)

    words = response.split()

    if not all(word[0].isupper() for word
        in words):
        return False

    scrabble_scores = [scrabble_score(
        word) for word in words]
    than the previous
    return all(current > prev for prev,
        current in zip(scrabble_scores,
        scrabble_scores[1:]))</pre> |

Table 7: Examples of AutoIF's data formats with multiple constraints.

# J   The illustration of FollowBench's Test Cases

The motivation of AUTOIF is to generalize to more complex instruction-following abilities by using a scalable and verifiable method through simple handwritten instructions. Therefore, we present the test examples from the complex instruction-following evaluation set Followbench we assessed. Followbench evaluates six dimensions, with each instruction having five levels of difficulty and comprising a series of integrated tasks. Below are three features of Followbench.

**Six Dimensions's Tasks of Followbench**: All constraints being evaluated for instruction following under the combination of various integrated tasks

1.  Content Constraint: Data-to-Text Generation, Document-Level Event Argument Extraction, Document-Level Named Entity Recognition, Text Generation with Language Constraints, Open-ended Question Answering

2. Situation: Suggestion Generation, Role-playing, Complex Situation Reasoning

3. Style: Open-ended Question Answering

4. Format: Text-to-Table Generation, Open-ended Question Answering

5. Example: 40 diverse NLP tasks

6. Mixed: Text Editing, Summarization, Machine Translation, Story Generation

**Examples of Constraints in Six Dimensions**: Each instruction's complexity cannot be resolved solely through surface semantics or 1-to-1 translation.

| Category | Test Case Description |
|---|---|
| **Content** | What, according to Milton Friedman, is the role of a business in society? Additionally, analyze its influence on ethical standards in society and identify one possible repercussion on relationships within the community. Please strengthen your argument with one relevant case study and its implications, along with citing one expert opinion or statistical data to support your viewpoint. |
| **Mixed Prompt** | Lost, found vodka, drank to forget. According to the above prompt, write a four-sentence story that describes a man. However, the word "man" should not appear in the story. Please write using an introspective narrative tone. You should also describe something about the bad weather. |
| **Situation** | If yesterday is Christmas Eve of 1937, what would be the date four years, a month, two weeks and two days after today in MM/DD/YYYY? |
| **Style** | How did US states get their names? Pray, respond in the writing style of Shakespeare and the elegance of the Victorian era, whilst infusing a touch of humor into thy discourse. Furthermore, craft thy response with the ambiguity reminiscent of the oracles of ancient Greece, leaving room for pondering and interpretation. As thou writest, channel the conciseness and vigor of Hemingway in thine articulation. |
| **Example** | Robert just called in and had some more details. He talked to Gay again. Sunny is OK, walked away from the wreck. It totaled her car. The airbag did not inflate so she was very lucky not to be hurt. He will report more when he gets there. Randy J. |
| **Format** | To enhance your time management skills, could you devise a method incorporating a mind map and featuring a touch of alliteration in the suggestion, ensuring your answer must follow the above suggestions. |

**Examples of Five Difficulty Levels:** For one constraint, the sentence's semantic structure greatly altered at higher levels:

Similarly, IFEval is a complex instruction evaluation combining multiple instructions and remains a core benchmark for foundational model instruction adherence [4].

---

[4] `https://github.com/google-research/google-research/blob/master/instruction_following_eval`

| Difficulty | Test Case Description |
|---|---|
| Level 1 | Identify one category from the list below for the input text, and also infer the sentiment (positive, neutral, or negative) conveyed in the text. Your options for the category are - company, educational institution, artist, athlete, office holder, means of transportation, building, natural place, village, animal, plant, album, film, or written work. Michael DenDekker - Michael G. DenDekker (born July 11, 1961) is an assemblyman for the state of New York's 34th district which includes the neighborhoods of Woodside, Jackson Heights, and East Elmhurst, all in the borough/county of Queens. |
| Level 2 | Identify one category and the sentiment conveyed (positive, neutral, or negative) in the input text, as well as conduct a named entity recognition task to locate and highlight the important entities present. You can choose the category from the following: company, educational institution, artist, athlete, office holder, means of transportation, building, natural place, village, animal, plant, album, film, or written work. Michael DenDekker - Michael G. DenDekker (born July 11, 1961) is an assemblyman for the state of New York's 34th district which includes the neighborhoods of Woodside, Jackson Heights, and East Elmhurst, all in the borough/county of Queens. |
| Level 3 | Analyze the provided text to pinpoint a category and the sentiment (positive, neutral, or negative) it emanates. Additionally, perform named entity recognition to emphasize notable entities and also identify the core topic discussed. Select the category from this array: company, educational institution, artist, athlete, office holder, means of transportation, building, natural place, village, animal, plant, album, film, or written work. Michael DenDekker - Michael G. DenDekker (born July 11, 1961) is an assemblyman for the state of New York's 34th district which includes the neighborhoods of Woodside, Jackson Heights, and East Elmhurst, all in the borough/county of Queens. |
| Level 4 | Analyze the supplied text to discern a category and the sentiment it conveys (positive, neutral, or negative). Furthermore, carry out named entity recognition to highlight significant entities and determine the main theme being discussed. In addition, perform keyword extraction to underline notable terms. Choose the category from this array: company, educational institution, artist, athlete, office holder, means of transportation, building, natural place, village, animal, plant, album, film, or written work. Michael DenDekker - Michael G. DenDekker (born July 11, 1961) is an assemblyman for the state of New York's 34th district which includes the neighborhoods of Woodside, Jackson Heights, and East Elmhurst, all in the borough/county of Queens. |
| Level 5 | Analyze the provided text to ascertain both the category and the sentiment (positive, neutral, or negative) it embodies. Additionally, conduct named entity recognition to emphasize important entities and establish the central theme. Moreover, undertake keyword extraction to mark prominent words, and engage in coreference resolution to identify references of the same entity within the text. Select the category from this array: company, educational institution, artist, athlete, office holder, means of transportation, building, natural place, village, animal, plant, album, film, or written work. Michael DenDekker - Michael G. DenDekker (born July 11, 1961) is an assemblyman for the state of New York's 34th district which includes the neighborhoods of Woodside, Jackson Heights, and East Elmhurst, all in the borough/county of Queens. |

Therefore, our cases and responses prove that the instruction following tasks are highly challenging, assessing the comprehensive capabilities of LLMs.

# K    MORE EXPERIMENT RESULTS OF AUTOIF

## K.1    VALIDATION IN LONG CONTEXT INSTRUCTION-FOLLOWING SCENARIO

To validate the generalization of AUTOIF in the fields of RAG and long windows, we conduct verification experiments on the FollowRAG benchmark (Dong et al., 2024a). As shown in Table 8, AUTOIF still shows significant improvements in long text scenarios, which further validates the effectiveness of our method in real-world challenging instruction-following contexts.

| Model | NQ (IF) | NQ (RAG) | NQ (AVG) | TQ (IF) | TQ (RAG) | TQ (AVG) |
|---|---|---|---|---|---|---|
| Llama3-8B-SFT | 15.7 | 59.5 | 37.6 | 15.0 | 76.5 | 45.7 |
| Llama3-8B (AutoIF) | 41.3 | 62.4 | 51.9 | 40.3 | 77.6 | 60.0 |

Table 8: Performance comparison of models on FollowRAG NQ and TQ benchmarks. Llama3-8B-SFT represents Llama3-8B finetuned on ShareGPT dataset and train set of NQ and TQ.

### K.2 MORE SETTINGS ON LOW RESOURCE SCENARIO

To validate the generalization of AUTOIF in scenarios with lighter resource consumption, we conduct an experiment using Llama3-8B-instruct for self-alignment with Llama3-8B-base, which can be effectively deployed using just one GPU. Additionally, to further challenge AUTOIF's potential in more demanding scenarios, we designed a **Weak-to-Strong** setup, enhancing Qwen2-7B with Qwen2-3B-instruct. This setup also requires only one GPU for effective deployment. As shown in Table 9, in both low-resource settings, AUTOIF consistently demonstrated stable improvements, highlighting its effectiveness.

| Method | Pr. (strict) | Pr. (L) | Ins. (S) | Ins. (L) | FollowBench (Avg.) |
|---|---|---|---|---|---|
| **Supervision Model: Qwen2-3B** | | | | | |
| Qwen2-7B-base | 37.7 | 43.6 | 49.4 | 53.4 | 52.3 |
| Qwen2-7B (ShareGPT) | 30.9 | 33.5 | 42.4 | 45.2 | 38.1 |
| Qwen2-7B (AutoIF) | 40.3 | 46.0 | 53.5 | 56.8 | 53.0 |
| **Supervision Model: Llama3-8B** | | | | | |
| Llama3-8B-base | 24.6 | 26.1 | 38.1 | 39.7 | 11.6 |
| Llama3-8B (ShareGPT) | 23.7 | 26.4 | 33.8 | 37.1 | 38.1 |
| Llama3-8B (AutoIF) | 32.5 | 37.7 | 43.3 | 49.2 | 44.2 |

Table 9: The weak to strong and self alignment setup on low resource scenario.

## L DISCUSSION ON CODE EXECUTION WORKS

Recent advancements in code generation and verification have produced several effective approaches. RLTF (Liu et al., 2023) generates data in real-time during training, utilizing multi-granularity unit test feedback to identify specific code errors, which helps improve code quality.

LEVER (Ni et al., 2023) enhances this process by training a verifier that assesses the correctness of programs generated by large language models (LLMs). It evaluates the generated code based on natural language inputs, execution results, and reorders candidates using a combined score of verification and LLM probability, ensuring optimal solutions.

Dotamath (Li et al., 2024a) tackles complex mathematical tasks by decomposing them into simpler logical subtasks, leveraging code to solve these subtasks, obtaining fine-grained feedback from the code interpreter, and engaging in self-reflection and correction.

ODEX (Wang et al., 2023b) introduces the first open-domain dataset for execution-based natural language to Python code generation, featuring 945 natural language-code pairs across 79 libraries and 1,707 manually written test cases for validation. This dataset is vital for training robust models in diverse programming contexts.

Lastly, Self-OSS-Instruct (Lozhkov et al., 2024) leverages context learning to enable the StarCoder2 model to autonomously generate diverse programming instructions from seed code snippets. This includes concept extraction and instruction generation, fostering a self-sufficient learning environment.

Collectively, these works highlight the importance of real-time feedback, verification mechanisms, comprehensive datasets, and self-learning strategies in enhancing the quality and reliability of code generation.

## M FUTURE WORK

AUTOIF, which first transforms instruction-following alignment into automatically code verification, requiring LLMs to generate instructions, corresponding verification code, and unit test samples for cross-validation. In the future, we find that constructing and verifying high-level semantic instructions (such as those with emotional or creative elements) is a key direction for enhancing the LLM alignment with human instruction following. Specifically, we believe there are several optimization avenues for AUTOIF to better accommodate high-level semantics:

- **Handwritten prompts:** We can consider fine-grained emotional differences in the prompts by handwriting instructions that allow for nuanced distinctions.

- **Instruction rewriting phase:** We can establish creative principles (e.g., for an emotional assistant, qualities like humor and empathy) and allow humans to iteratively optimize these principles based on the quality of generated outputs from small batches, potentially using instruction evolution techniques like AutoEval instructions [1]. Principle of LLM verification: Inspired by CAI [2], we also need to incorporate fine-grained emotional differences in the verification prompts during the verification phase or use creative metrics for scoring, rather than solely focusing on instruction correctness, to overcome the limitations of executor-based verification that only addresses verifiable prompts.

- **Online/Offline DPO data construction:** For creative tasks, we should avoid using executor-based success rates to construct positive and negative samples. Instead, a combination of LLM verification scores and executor-based scores should be employed to balance correctness with higher-level emotional semantics.

