# OpenReview forum: "Self-play with Execution Feedback: Improving Instruction-following Capabilities of Large Language Models"
_ICLR.cc/2025/Conference — ICLR 2025 Spotlight_

### Official Review · Reviewer_5kXL · 2024-10-28

**Soundness:** 3
**Presentation:** 3
**Contribution:** 2
**Rating:** 6
**Confidence:** 4

**Summary:**

The authors propose AUTOIF, a training framework to improve instruction following capability via execution feedback. The authors demonstrate the efficacy of AUTOIF on four LLMs with benchmarks across various domains.

**Strengths:**

There are a few valuable contributions:
- The proposed AUTOIF shows that training with program-based validation can improve some instruction-following capabilities.
- The experiments are pretty extensive, where the authors benchmark the AUTOIF on four models with different sizes.
- The ablation studies are quite detailed, showing the optimal setups for AUTOIF.

**Weaknesses:**

There are several weaknesses to be addressed:
- The idea of using programs to validate instruction-following is not novel, as it has been used in IFEval. Moreover, training with execution feedback is also not new; it has been widely used by training code LLMs [1-3].
- Based on the proposed workflow, AUTOIF is hard to scale to complex instructions, where the queries are totally quantifiable. AUTOIF cannot easily work for semantically rich instructions, which cannot be evaluated via execution. In addition, real-world scenarios that can be validated via programs may require various dependencies and APIs. AUTOIF is not scalable under these practical cases.
- The instruction-following evaluations on IFBench and FollowBench lack practicability. These two benchmarks only contain quantifiable instructions, like word counts. The authors should consider more practical scenarios like the one in BigCodeBench [4] for the instruction-following evaluation.
- The current evaluation is English-only. The authors should evaluate their models on multilingual setups to demonstrate their generalisability.

[1] Le, H., Wang, Y., Gotmare, A. D., Savarese, S., & Hoi, S. C. H. (2022). Coderl: Mastering code generation through pretrained models and deep reinforcement learning. Advances in Neural Information Processing Systems, 35, 21314-21328.

[2] Zheng, T., Zhang, G., Shen, T., Liu, X., Lin, B. Y., Fu, J., ... & Yue, X. (2024). Opencodeinterpreter: Integrating code generation with execution and refinement. arXiv preprint arXiv:2402.14658.

[3] Chen, B., Zhang, F., Nguyen, A., Zan, D., Lin, Z., Lou, J. G., & Chen, W. CodeT: Code Generation with Generated Tests. In The Eleventh International Conference on Learning Representations.

[4] Zhuo, T. Y., Vu, M. C., Chim, J., Hu, H., Yu, W., Widyasari, R., ... & Von Werra, L. (2024). Bigcodebench: Benchmarking code generation with diverse function calls and complex instructions. arXiv preprint arXiv:2406.15877.

**Questions:**

Apart from the mentioned weaknesses, there are some additional questions:
- How do the accuracy rate thresholds in the verification steps affect the model performance?
- How does RAILF+Online DPO (e.g., OAIF [1]) compare to AUTOIF?

[1] Guo, S., Zhang, B., Liu, T., Liu, T., Khalman, M., Llinares, F., ... & Blondel, M. (2024). Direct language model alignment from online ai feedback. arXiv preprint arXiv:2402.04792.

---

> ### Author Response · Authors · 2024-11-19
> **Response (1/3)**
>
> Thanks for your helpful comments and we are glad that you acknowledged the strengths of our work! Below we will address your separate concerns.
>
> ---
>
> ### W1: Using programs to verify instructions following has already been implemented in IFEval, Moreover, using execution feedback for training is not novel, as it has been widely applied in training code LLMs [1-3].
>
>
> Thank you for your suggestion. We would like to clarify the distinctions between our work and the three related Code LLM efforts as follows:
>
> - **From the perspective of code generation goals**: References [1-3] focus on code generation tasks aimed at producing more reliable code, while AutoIF's goal is to enhance the general instruction-following capabilities of LLMs under complex natural language instructions, with code used solely to verify whether the model's responses meet the criteria.
>
> - **From a technical standpoint**: CodeRL combines pre-trained language models with deep reinforcement learning, focusing on assessing the functional correctness of generated programs through a critic network. Opencodeinterpreter integrates code generation, execution, and iterative optimization. Both of these works are significantly different from AutoIF. While CodeT shares some similarities with AutoIF in generating test cases, it provides tests for generated code, whereas AutoIF uses program execution to validate the model's instruction-following ability, representing a fundamental difference.
>
> Additionally, AutoIF introduces Executor Feedback along with model scoring and recall verification mechanisms. Specifically, instruction-following tasks are more diverse and semantically rich, so AutoIF not only relies on code execution results but also utilizes a scoring mechanism to validate generated responses from multiple angles, enhancing data reliability and quality. This further differentiates it from the IFEval dataset, which focuses solely on rule-based verification instructions. We sincerely hope that reviewers will appreciate the comprehensiveness of our innovations in verification. The ablation study (Table 4) also demonstrates that these mechanisms complement each other.
>
> ---
>
> ### W2 (1): AutoIF struggles to scale to complex instructions and has difficulty handling semantically rich instructions that cannot be evaluated through execution.
>
>
> Thank you for your comment. The motivation behind AUTOIF is to generalize complex instruction-following abilities using a scalable, verifiable method through simple handwritten instructions. We will respond in two parts:
>
> **Generalization of Semantically Rich Instructions**: Actually, the datasets evaluated in Tables 1 and 2 contain semantically rich instructions:
>
> 1. **FollowBench**: This dataset evaluates instruction following across over 50 integrated NLP tasks with six difficulty levels. Many instructions are unverifiable and require GPT-4 scoring, as detailed in the case examples in Appendix 6.
>
> 2. **MT-Bench**: A comprehensive natural instruction evaluation set that focuses on challenging questions to differentiate model capabilities, consisting of 80 high-quality multi-turn dialogue questions across eight use cases: writing, role-playing, information extraction, reasoning, mathematics, coding, knowledge I (STEM), and knowledge II (humanities/social sciences).
>
> 3. **InfoBench**: This dataset further validates our OOD instruction-following capabilities and lacks verifiable instructions, with all evaluations requiring external model scoring.
>
> 4. **Arena Hard**: A highly challenging evaluation set aimed at realistically assessing comprehensive chatbot performance.
>
> We believe these integrated tasks effectively validate AUTOIF's generalization in real-world applications.
>
> **Potential Optimization Approaches**: Furthermore, we are open to discussing optimization strategies for the more high-level semantic instruction constraint. Here are a few ideas:
> 1. Establish principles for style, such as defining "politeness," which might include honorifics and exclude offensive language. Verification methods could extend beyond executor feedback.
> 2. Clearly define style types and manually create style-related seed instances, using model pairwise comparisons for quality validation.
>
> These are preliminary thoughts and not the main focus of this paper, but we are willing to explore further style optimization as future work.

---

> ### Author Response · Authors · 2024-11-19
> **Response (2/3)**
>
> ### W2 (2): Real-world scenarios for AutoIF program verification may require various dependencies and APIs.
>
> Thanks for the advice! Addressing concerns about APIs, the models reported in Table 1 are deployable open-source models, Llama3-70B and Qwen2-70B, used as supervision models, rather than GPT-4. To further alleviate your concerns, we present the self-alignment using Llama3-7B as the supervision model, as well as the setup where Qwen2-3B-instruct enhances Qwen2-7B. These supervision models can be lightweight **deployed using a single GPU**.
>
>
> |Method        | Pr. (strict) | Pr. (L) | Ins. (S) | Ins. (L) | FollowBench (AVG) |
> |----------------------------|---------------|---------|----------|----------|------------------|
> | **Supervision Model: Qwen2-3B-Instruct**      |               |         |          |          |                  |
> | Qwen2-7B-base              | 37.7          | 43.6    | 49.4     | 53.4     | 52.3             |
> | Qwen2-7B (ShareGPT)       | 30.9          | 33.5    | 42.4     | 45.2     | 38.1             |
> | Qwen2-7B (AutoIF)         | 40.3          | 46.0    | 53.5     | 56.8     | 53.0             |
> | **Supervision Model: Llama3-8B-Instruct**     |               |         |          |          |                  |
> | Llama3-8B-base             | 24.6          | 26.1    | 38.1     | 39.7     | 11.6             |
> | Llama3-8B (ShareGPT)      | 23.7          | 26.4    | 33.8     | 37.1     | 38.1             |
> | Llama3-8B (AutoIF)        | 32.5          | 37.7    | 43.3     | 49.2     | 44.2             |
>
> Additionally, as detailed in our supplementary materials, all compiler execution processes can be completed with **a single command on the CPU**. We hope these results alleviate your concerns regarding computational resources!
>
> ---
>
> ### W3: IFEval and FollowBench lack practicality, as they only include verifiable instructions like word count. The authors should consider more realistic scenarios, such as those in BigCodeBench [4].
>
> Thank you for your suggestion. First, we think reviewer may be underestimating the difficulty of FollowBench. As shown in appendix J, FollowBench has the following two characteristics:
>
>
> - **Six Dimensions's Tasks of Followbench:** All constraints being evaluated for instruction following under the combination of various integrated tasks
>
> ```
> 1. Content Constraint: Data-to-Text Generation, Document-Level Event Argument Extraction, Document-Level Named Entity Recognition, Text Generation with Language Constraints, Open-ended Question Answering
> 2. Situation: Suggestion Generation, Role-playing, Complex Situation Reasoning
> 3. Style: Open-ended Question Answering
> 4. Format: Text-to-Table Generation, Open-ended Question Answering
> 5. Example: 40 diverse NLP tasks
> 6. Mixed: Text Editing, Summarization, Machine Translation, Story Generation
> ```
>
> - **Examples of Constraints in Six Dimensions:** Each instruction’s complexity cannot be resolved solely through surface semantics or 1-to-1 translation. Detailed cases are listed in the Appendix J (Page 23-24).
>
> Additionally, as mentioned in our W(2) response, we also evaluate AUTOIF on more practical scenarios in Table 2, including:
>
> 1. InfoBench：OOD instruction following benchmark
> 2. MT-Bench：More natural and general instruction following capability
> 3. Arena Hard (Chatbot Arena)：Realistically challenging comprehensive chatbot evaluation
>
> This paper primarily focuses on evaluating AUTOIF's general instruction-following capabilities. Evaluating instructions in specific domains, such as code and mathematics is not the core focus of this work;  However, we are will to expanding evaluations in code-related areas like BigCodeBench in the future.
>
> ---
>
> ### W4: AUTOIF should be evaluated in a multilingual context to demonstrate its generalization ability.
>
> Thank you for your insightful suggestion. In fact, based on our simple and efficient workflow, AUTOIF framework is completely language-agnostic. Therefore, we believe that extending it to multiple languages requires only changing the seed instruction to the specific instruction in another language, along with selecting a supervision model with additional capabilities. All other processes can be reused. Due to the tight rebuttal timeline, we are very willing to follow your advice and include more multilingual benchmarks in our future work to enhance its generalization ability.

---

> ### Author Response · Authors · 2024-11-19
> **Response (3/3)**
>
> ### Q1: How do the accuracy rate thresholds in the verification steps affect the model performance?
>
> Thank you for your suggestion. In fact, in the left two graphs of Figure 4, we have explored the impact of increasing the accuracy threshold in the verification step from both the instruction and query perspectives, as well as its effect on the final data volume.
>
> Here are our core findings in Figure4: As the pass rate threshold increases, the amount of SFT data decreases at the instruction level, while model performance consistently improves. This indicates that the quality of instructions is a crucial factor influencing instruction-following (IF) performance. At the query level, the amount of SFT data also decreases with higher pass rate thresholds. Notably, performance peaks at a pass rate of 0.8 and declines beyond 1. This observation aligns with our expectations, highlighting a trade-off between data quality and quantity as the threshold increases. We hope these can address your concerns.
>
> ---
>
> ### Q2: How does RAILF+Online DPO (e.g., OAIF) compare to AUTOIF?
>
> Thank you for your insightful suggestion. After carefully reviewing the OAIF work you pointed out, both AutoIF and OAIF employ online DPO training strategies to enhance model performance. The key difference is that AutoIF focuses on using code validation to improve the model's ability to follow complex instructions, while OAIF utilizes LLM annotators to label positive and negative samples, aligning with the model's preferences. The core motivation of this paper is to emphasize the automation and scalability of enhancing instruction adherence with minimal human effort. Therefore, OAIF is not suitable as a baseline for comparison in our study.
>
> ---
>
> Finally, we sincerely hope that our response addresses your concerns, and we once again thank you for your thorough review!

---

> ### Comment · Reviewer_5kXL · 2024-11-19
>
> Thank you for the response. It addressed some of my concerns and helped me understand the work better.
>
> However, regarding OAIF, I would like to respectfully argue that RAILF+Online DPO still works in the case of instruction following. Considering that LLMs can evaluate whether generated responses are aligned with the instructions, it should be one of the baselines. As discussed, such methods should work well with style- and semantic-related instructions.

---

> > ### Author Response · Authors · 2024-11-22
> > **Response to Reviewer 5kXL (Follow-up)**
> >
> > Thank you for your timely response! To further address reviewer's concern, we are willing to adopt OAIF as our baseline for experimental validation.
> >
> > Specifically, we have carefully reviewed and followed the OAIF framework, using our Strong-to-Weak setting. We utilize Llama3-70B-Instruct as the supervision model, first synthesizing the SFT dataset according to the AUTOIF framework, and then supervised fine-tuning the Llama3-8B-base model. After each round of fine-tuning, we sample two responses from the current SFT model and let the supervision model choose the preferred one, providing online feedback. Notably, this process can be iterated.
> >
> > | Method                           | Pr. (S) | Pr. (L) | Ins. (S) | Ins. (L) |
> > |--------------------------------|---------|---------|----------|----------|
> > | AUTOIF-SFT（Llama3-8B）       | 28.7    | 40.3    | 41.4     | 52.2     |
> > | + 1 round OAIF                 | 27.5    | 41.0    | 41.0     | 52.9     |
> > | + 2 round OAIF                 | 28.2    | 41.8    | 41.6     | 53.5     |
> > | + 1 round AUTOIF Online-DPO    | 27.9    | 41.6    | 40.5     | 54.1     |
> > | + 2 round AUTOIF Online-DPO    | 28.8    | 43.1    | 42.2     | 56.0     |
> >
> > Due to limited time and computational resources, we conduct two rounds of online DPO for both OAIF and AUTOIF on IF-Eval. The results show that both methods experience a decline in strict metrics for Prompt and loose level during the first round, but this issue was significantly alleviate after the second round. In comparison, AUTOIF demonstrate more significant improvements in each optimization round than OAIF. It is worth mentioning that the online DPO data for AUTOIF is automatically compiled and validated using a verification function generated during the synthesis phase, relying solely on CPU resources, which allows for faster annotation. In contrast, the OAIF process incurs additional inference computational overhead. This difference highlights the inherent advantages of the AUTOIF framework in terms of high performance and low computational consumption.
> >
> > In the future, we will supplement OAIF experiments on other datasets (such as FollowBench and InfoBench) and add other RAILF+Online DPO baselines to enhance completeness.
> >
> > We hope these results will alleviate your concerns, and thank you once again for your feedback!

---

> > > ### Comment · Reviewer_5kXL · 2024-11-22
> > >
> > > Thank you for the timely results! I've raised my scores accordingly.

---

### Official Review · Reviewer_TNGe · 2024-11-01

**Soundness:** 3
**Presentation:** 2
**Contribution:** 3
**Rating:** 8
**Confidence:** 3

**Summary:**

The paper proposes AUTOIF, an automated, scalable, and reliable method designed to enhance the instruction-following capabilities of LLMs, using automatic code verification.

**Strengths:**

There is a detailed and structured pipeline of AutoIF which showcases enhanced results over multiple training strategy.

**Weaknesses:**

However, examples on failed cases and illustration of these categories of consistent failure would help understand scope of next steps and areas of improvement that has been brought about by the method. Along with the limited domain of seed instructions in terms of string manipulation only, which could be diversified.

**Questions:**

- Figure 2, improve consistency of terms in figure with 'acc' and 'Acc' being used alternatively, and the sequence of verification and func set being inverted at start.
- An example case of how a sample develops in the AutoIF pipeline would help illustrate the working even better.
- Line 215, 232, provide reasoning and example of failing cases of why accuracy is lower. similarly for Section 4.1 talks about main results but is limited to quantitative numbers and not examples of illustrating failed cases.
- Also, most seed instruction showcase restriction over string manipulation domain only. Ensuring type of word to be present, limiting length of output and so on. There is no showcasing of diverse, novel instructions.
- line 319, what is the value of K, for which the results have been sampled from?

---

> ### Author Response · Authors · 2024-11-23
> **Response for Reviewer TNGe**
>
> Thanks for your helpful comments and we are glad that you acknowledged the strengths of our work! Below we will address your separate concerns.
>
> ---
>
> ### Weakness: Failed cases will help understand the scope of the next steps and areas for improvement. Additionally, the seed instructions are limited to the domain of string manipulation, which could also be diversified.
>
> Thank you for your insightful suggestions. As you mentioned, failed cases can indeed reveal many areas for improvement. From the lower pass rates in cross-validation samples, we found that constructing and verifying high-level semantic instructions (such as those with emotional or creative elements) is a key direction for enhancing the LLM alignment with human instruction following. Specifically, we believe there are several optimization avenues for AUTOIF to better accommodate high-level semantics:
>
>
> - Handwritten prompts: We can consider fine-grained emotional differences in the prompts by handwriting instructions that allow for nuanced distinctions.
> - Instruction rewriting phase: We can establish creative principles (e.g., for an emotional assistant, qualities like humor and empathy) and allow humans to iteratively optimize these principles based on the quality of generated outputs from small batches, potentially using instruction evolution techniques like AutoEval instructions [1].
> - Principle of LLM verification: Inspired by CAI [2], we also need to incorporate fine-grained emotional differences in the verification prompts during the verification phase or use creative metrics for scoring, rather than solely focusing on instruction correctness, to overcome the limitations of executor-based verification that only addresses verifiable prompts.
> - Online/Offline DPO data construction: For creative tasks, we should avoid using executor-based success rates to construct positive and negative samples. Instead, a combination of LLM verification scores and executor-based scores should be employed to balance correctness with higher-level emotional semantics.
>
> These are just some experiential thoughts and are not the main focus of this paper. We are open to addressing further optimization in future work
>
>
> ---
>
> ### Q1: Please improve consistency of terms in figure.
>
> Thank you for your correction! We will work on improving the consistency of terminology between the text and figures, as well as proofreading the entire paper.
>
> ---
>
> ### Q2 & Q3: An example case of how a sample develops in the AutoIF pipeline and providing reasoning and examples for cases with lower accuracy.
>
>
> ---
>
> Thank you very much for your comments. In Appendix, Tables 6 and 7 present case examples of AutoIF, including the query, response, and verification function, to help readers understand our process. In the supplementary materials, we provide a series of JSONL files in the sample_data folder that illustrates the development process of our samples within the AutoIF framework, particularly the intermediate data results in "dpo_query_w_funcs.jsonl" for reviewer reference.
>
> In the revised version, we plan to include an image that details the process from a seed instruction to the final constructed sample, highlighting the verification at each stage. Additionally, we will include more detailed error cases to enhance the presentation of our paper.
>
> ---
>
> ### Q4: Most seed instruction showcase restriction over string manipulation domain only.
>
> That is good point! Our motivation is to synthesize high-quality instruction-following alignment datasets through fewer than 100 simple instructions in an automated, verifiable, and scalable process; therefore, our paper did not specifically optimize the seed instruction set. To increase the diversity of instructions, we could manually introduce higher-level semantic instructions or domain-specific instructions during the design phase, followed by targeted optimization using AutoIF.
>
> ---
>
> ### Q5: line 319, what is the value of K, for which the results have been sampled from?
>
> Thank you for your advice. Our value of K is 8. For details, please refer to the dpo_query_eval_score_results.jsonl file in the sample_data folder of our supplementary materials, which displays eight sample entries from the data. These samples were extracted from our AUTOIF SFT training set. We will incorporate this information further into our implementation details.
>
> ---
>
> ### Reference:
>
> [1] Automatic Instruction Evolving for Large Language Models
>
> [2] Constitutional AI: Harmlessness from AI Feedback

---

### Official Review · Reviewer_BmDh · 2024-11-03

**Soundness:** 2
**Presentation:** 2
**Contribution:** 2
**Rating:** 8
**Confidence:** 4

**Summary:**

The paper proposes AutoIF, a scalable method for generating high-quality instruction-following data for training LLMs without manual annotation. It leverages code execution to verify response quality. LLMs are used to generate instructions, produce the corresponding code, and create unit test samples for cross-validation. Execution-based rejection sampling produces data for SFT and RLHF. Applied to Qwen2 and Llama3, AutoIF enhances performance across three training algorithms (SFT, offline DPO, online DPO), achieving over 90% accuracy in IFEval’s loose instruction accuracy benchmark without sacrificing general, math, or coding abilities, highlighting its effectiveness in alignment and generalization.

**Strengths:**

1. The technique is intuitive. IFEval was constructed using verifiable instructions to test an LLM's instruction-following ability, while this paper generates verifiable code to evaluate the instruction-response alignment.
2. AutoIF proves to be effective in the self-alignment case, where the same model is used for instruction generation and verification.
3. The evaluation covers a wide range of benchmarks and shows that AutoIF improves instruction following significantly while preserving the other general capabilities.
4. AutoIF naturally generates pair-wise preference data and can be applied to DPO, offline or online.

**Weaknesses:**

1. In the self-alignment setting, the evaluation is only conducted on the two >70B models. It is unclear whether the same conclusion applies to weaker base models with less than 10B parameters.
2. There are strong-to-weak and self-alignment settings, so a weak-to-strong setting should logically be included. There’s no proper justification for leaving it out.
3. There's a missing baseline where responses are validated through the LLM itself without code. Without the baseline, it's unclear how valuable the verification function and test cases are.

**Questions:**

1. See the weaknesses above.
2. Line 107: `we validate AUTOIF’s effectiveness in both "Self Alignment" and "Weak to Strong"`. typo?
3. Will the self-alignment setting still work for smaller models like Llama3-8B?
4. What are the main technical differences from StarCoder2-Instruct, which also used the base model itself to generate instructions, code, and test cases for validation?
5. Although the IFEval and FollowBench results are impressive, these benchmarks are specifically designed to test natural language instruction following. Can improved IFEval performance be linked to more practical tasks? For instance, while benchmarks like GSM8K and HumanEval may not show additional gains, BigCodeBench / LiveCodeBench / LeetCode include complex instruction-following cases, which should ideally lead to improvement.
6. Are there analyses on how reliable the generated code and tests are? Either or both of them may be just wrong.

If my concerns are properly addressed, I will consider raising the score.

---

> ### Author Response · Authors · 2024-11-19
> **Response (1/2)**
>
> Thanks for your helpful comments and we are glad that you acknowledged the strengths of our work! Below we will address your concerns point by point:
>
> ---
>
> ### W1 & Q3:  It is unclear whether the self-alignment setting applies to weaker base models with less than 10B params (e.g., Llama3-8B).
>
> Thank you for your comments. We have evaluated the Llama3-8B-instruct version as a supervision model for self-alignment with the Llama3 8B base model. The experimental results are as follows:
>
> | Method                 | Pr. (strict) | Pr. (L) | Ins. (S) | Ins. (L) | FollowBench(AVG.)  |
> |-----------------------|---------------|---------|----------|----------|------|
> | Llama3-8B-base        | 24.6         | 26.1    | 38.1     | 39.7     | 11.6 |
> | Llama3-8B (ShareGPT)  | 23.7         | 26.4    | 33.8     | 37.1     | 38.1 |
> | Llama3-8B (AutoIF)    | 32.5         | 37.7    | 43.3     | 49.2     | 44.2 |
>
>
> The left four columns represent the results of IFEval, while the right side shows the results of FollowBench SSR AVG.
>
> Results indicate that compared to the SFT version of the Llama-8B (ShareGPT), Llama3 (AutoIF) achieves a significant improvement of over 12% on IFEval(Loose Instruction ACC.), while maintaining a 6% improvement on FollowBench. This validates the effectiveness of AutoIF in the self-alignment setup with the weaker model (only 7B). We will incorporate these results and analyses into the revised version.
>
>
> ---
>
> ### W2：Lack of weak-to-strong setup.
>
>
> Thanks for the advice! Honestly, we believe that the weak-to-strong setup is indeed quite demanding and challenging, as the data synthesis process of AutoIF comprehensively tests the supervision model's capabilities in instruction rewriting, code generation, and quality scoring. Further focusing on real-world scenarios, we rarely use weak models to enhance the instruction-following capabilities of strong models, which is somewhat counterintuitive. Given these reasons, we do not conduct experiments in the main text.
>
>
>
> To improve the completeness of the AutoIF setup and further address the reviewer's concern, we align the model series and ultimately chose Qwen2 3B-Instruct to perform weak-to-strong alignment with the Qwen2 7B base model. The experimental results are as follows:
>
>
> | Model                 | Pr. (strict) | Pr. (L) | Ins. (S) | Ins. (L) | FollowBench (AVG) |
> |-----------------------|---------------|---------|----------|----------|------------------|
> | Qwen2-7B-base         | 37.7         | 43.6    | 49.4     | 53.4     | 52.3             |
> | Qwen2-7B (ShareGPT)   | 30.9         | 33.5    | 42.4     | 45.2     | 38.1             |
> | Qwen2-7B (AutoIF)     | 40.3         | 46.0    | 53.5     | 56.8     | 53.0             |
>
>
> Results show that the weak-to-strong setup can also achieve stable improvements. We attribute this gain to the cross-capability benefits derived from code generation and instruction rewriting by the weak model, demonstrating that the AutoIF framework possesses strong generalization. We will also include these experimental results and analyses in the revised version.
>
>
> ---
>
> ### W3: Missing baseline where responses are validated through the LLM itself without code
>
>
> Thank you for your suggestion. As shown in Table 4 of the ablation study, the setting for ablation of cross verification（executor-based flitering） refers to the synthetic data of AutoIF, where we retain only the final stage of LLM query quality verification and do not conduct the executor feedback cross-validation process. This is essentially equivalent to the baseline process you mentioned, as in this setting, the data originally filtered by the code validation will ultimately undergo LLM Query Quality Verification.
>
> | **Model**                             | Pr. (L) | Ins. (L) | FollowBench (SSR) |
> |---------------------------------------|------------|------------------|-------------------|
> | Qwen2-7B-SFT-Online DPO   | 46.6      | 57.9             | 56.6              |
> | *w/o* Quality Verification             | -1.4      | -2.4             | -1.3              |
>
>
> As noted in the analysis: "The most significant performance drop occurs when the Cross Verification of instructions is removed, underscoring its importance over query quality verification. This verifies that a high-quality instruction set is fundamental to the AutoIF process." We believe this result validates the value of the verification function and test cases in AutoIF.
>
> ---
>
> ### Q2: Typos in line 107.
>
>
> Thank you for your correction; it should be the "Strong-to-Weak" setting. However, as we verified the "Weak to Strong" setting in W2, we are happy to incorporate this aspect into both the contributions and the experiments.

---

> ### Author Response · Authors · 2024-11-19
> **Response (2/2)**
>
> ### Q4：Please clarify the main technical differences with StarCoder2-Instruct.
>
> This is a good point. While StarCoder2-Instruct utilizes a base model for code and test case generation, AutoIF introduces significant innovations and optimizations in four key areas:
>
> 1. **Task and Motivation**:
>    - StarCoder2 enhances task completion rates for code generation tasks. AutoIF focuses on **cross abilities enhancement**, improving general instruction adherence through diverse capabilities like code generation and Instruction rewriting.
>
> 2. **Verification Mechanism**:
>    - StarCoder2-Instruct filters responses by executing generated test cases in a sandbox, essentially sampling outputs. In contrast, AutoIF employs **Executor Feedback, Back-translation Verification, and LLM Quality Verification.**, validating generated responses through execution results, NLI Models and the scoring mechanism based on LLM, enhancing both reliability and quality.
>
> 3. **Scalability and Versatility**:
>    - While StarCoder2 excels in SFT dataset generation, it primarily targets programming task responses. AUTOIF automates instruction-response pair generation and validation across various training algorithms (SFT, Offline DPO, Online DPO), offering greater scalability.
>
> 4. **Experimental Aspects**:
>    - StarCoder emphasizes coding task performance. AUTOIF enhances general instruction-following capabilities and validates its plug-and-play framework across domains without conflicts math, code and general domain. Its varied settings and diverse supervision models (GPT-4, LLaMA, Qwen series) further broaden its applicability.
>
>
> ---
>
> ### Q5: Can improved IFEval performance be linked to more practical tasks? GSM8K and HumanEval may not show additional gains.
>
> Thank you for your suggestion. In fact, we believe benchmarks evaluated in Tables 1 and 2 are all relevant to real-world tasks:
>
> - **FollowBench**: This assesses instruction-following across **over 50 integrated NLP tasks** with six difficulty levels. Most instructions cannot be validated and require GPT-4 for evaluation, as detailed in Appendix 6.
>
> - **MT-Bench**: A comprehensive evaluation set focusing on challenging questions to differentiate model capabilities, it includes 80 high-quality multi-turn dialogue questions across eight use cases: **writing, role-playing, information extraction, reasoning, mathematics, coding, STEM, andhumanities/social sciences**.
>
> - **InfoBench**: This dataset further validates our **out-of-distribution instruction-following capabilities**, also lacking verifiable instructions, with all evaluations requiring external model scoring.
>
> - **Arena Hard (win rate)**: One of the most challenging evaluation sets, aimed at **realistically assessing comprehensive chatbot performance**.
>
> Thus, the results in Tables 1 and 2 effectively validate the generalization of AutoIF in real-world tasks.
>
> Regarding the GSM8K and HumanEval results in Table 1, we consider there may be a misunderstanding for reviewer. As we mentioned in our contributions, we include general (MMLU, C-Eval), math (GSM8K), and coding (HumanEval) abilities to show that **improvements in instruction-following capabilities do not conflict with other abilities**. Therefore, we do not need to enhance these abilities, their stability without significant decline proves that AutoIF avoids performance conflicts, and it even achieves slight improvements.
>
> Moreover, we are willing to exploring further instruction-following validation in specialized code domains (BigCodeBench / LiveCodeBench / LeetCode) as future work.
>
> ---
>
> ### Q6：Reliability Analysis of Generated Code and Tests
>
> Thank you for your insightful suggestion. In Table 5, we discuss the relationship between the code capabilities of the supervision model and instruction-following abilities, presenting the pass rates of test cases. To enhance our experiments' completeness, we incorporated your suggestion and conducted a reliability analysis of the generated code and tests. We sample 10K instruction verification codes generated by GPT-4 and Qwen2-72B and evaluatetheir code execution rates. After excluding external factors (e.g., compiler environment bugs), we present the code execution rate results alongside Table 5 for analysis.
>
> | Model      | Code Execution Rate | Case Pass Rate | MBPP | IFEval |
> |------------|---------------------|----------------|------|--------|
> | GPT-4     | 76%                 | 28%            | 73.9 | 59.3   |
> | Qwen2-72B | 68%                 | 34%            | 87.5 | 44.7   |
>
> The experiments show that the **code execution rate is reasonable and positively correlated with the code capabilities of both models**. Notably, MBPP performance, code pass rate, and instruction-following capabilities are consistently positively correlated. This reinforces our conclusion on "Data Efficiency," where the **supervision model's coding ability significantly impacts data synthesis quality and final instruction-following capability**.

---

> > ### Comment · Reviewer_BmDh · 2024-11-19
> >
> > Thanks to the authors for the detailed response, which addressed many of my concerns. I’ve increased my score.

---

> > > ### Author Response · Authors · 2024-11-19
> > > **Thank you for Reviewer‘s Reply**
> > >
> > > Thank you very much for your prompt response! We will incorporate these results into the revised version and carefully proofread it.

---

### Official Review · Reviewer_7mrP · 2024-11-03

**Soundness:** 3
**Presentation:** 3
**Contribution:** 3
**Rating:** 8
**Confidence:** 4

**Summary:**

This paper presents AutoIF. The main idea of AutoIF is to generate constraint-following instruction data by generating validation functions that correspond to the constraints within the natural language instruction and running them to validate any prospective responses.

**Strengths:**

- This paper looks at an important problem of training LLMs to precisely follow instructions
- The proposed approach is novel and seems general to most scenarios. It can be used to improve the quality of instruction-following datasets.
- This paper comes with comprehensive end-to-end evaluations as well as controlled experiments, with good insights and reasonable explanations.

**Weaknesses:**

- Section 3.2: cross-verification can be weak as it essentially assumes test-function self-consistency as correctness. Not sure how reliable is self-consistency here as (i) bad tests can kill the right function and (ii) tests and functions can be broken all together result in corrupted data quality.

----

Other minor suggestions:

- Code execution has also been widely adopted in the code domain to generate test-assured code; it is generally good to discuss more of such work (not only just CodeRL; but also RLTF, LEVER, Self-OSS-Instruct, ODEX, etc.) to connect the general research context.
- L209: It might be more accurate to cite the use of rejection sampling with "Training a helpful and harmless assistant with reinforcement learning from human feedback" in addition to/instead of the Llama 2 paper.
- L210: what does "_{j=1}" mean? Do you mean "i=1"?

**Questions:**

- I am curious about counter-examples where programs cannot model the constraints in the instruction; and how to recognize such cases and delegate them to human experts.
- Section 3.2: How reliable is cross-validation of test cases and verification functions? Do you see cases where cross-consistent test and verifiers are wrong together?

---

> ### Author Response · Authors · 2024-11-23
> **Response (1/2)**
>
> Thanks for your helpful comments and we are glad that you acknowledged the strengths of our work! Below we will address your concerns point by point:
>
> ---
>
> ### Weakness & Q2: Cross-verification may be unreliable because it essentially assumes the correctness of the test function. This is due to (i) poor test samples potentially undermining a correct function, and (ii) the possibility that both the test and the function may be incorrect simultaneously, leading to low-quality samples.
>
> This is indeed a great question! I would like to address your concern from two aspects:
>
> **Effectiveness of Self-Consistency**: In fact, our cross-validation incorporates a self-validation process with four criteria （line 214-234）, and their combination makes this process more rigorous. As demonstrated in our ablation study, we show the impact of removing this cross-validation:
>
> | Model                                                     | Pr. (L) | Ins. (L) | FollowBench (SSR) |
> |-----------------------------------------------------------|---------|----------|-------------------|
> | Qwen2-7B (AutoIF)                                       | 46.6    | 57.9     | 56.6              |
>  Qwen2-7B w/o Cross Verification | 45.2    | 55.5     | 55.3              |
>
> It is evident that our self-consistency verification can significantly enhances the instruction following capability of LLMs.
>
> **Manual Error Rate Validation**: Furthermore, to alleviate your concerns, we manually checked 50 test cases after cross-validation, verifying the function and identifying any errors that occurred simultaneously.
>
>
> Metric | Test Cases | Verification Funcs | Both |
> |------------|--------------------|------|-------------|
> |      Error Rates       |    16%               |  24%    | 8%
>
> Statistics indicate that simultaneous errors of both test cases and verification funcs are relatively rare. This is because our motivation stems from automated synthesis, utilizing simple and verifiable instructions that ultimately generalize to complex, general instruction-following tasks, ensuring the overall process is reliable. Additionally, AUTOIF incorporates "LLM-based quality verification" and "NLI model-based back-translation verification", which we believe can effectively address any oversights in executor-based validation.
>
> ---
>
> ### Comment1: It is generally good to discuss more of code execution works.
>
>
> Thank you for your suggestions. We are happy to add a section in the appendix to discuss Code execution, including RLTF, LEVER, Self-OSS-Instruct, ODEX, and other related works mentioned by reviewers. Here is an overview of these works we summarized:
>
> - **RLTF** generates data in real-time during training to continuously update and optimize the model. It incorporates multi-granularity unit test feedback, using fine-grained signals to pinpoint specific errors in the code, guiding the model toward generating higher-quality code.
>
> - **LEVER** trains a verifier to assess whether programs generated by large language models (LLMs) are correct. The verifier's judgment is based on natural language input, the generated program, and its execution results. After generating a program, it is reordered based on the verification score combined with the LLM generation probability, marginalizing programs with the same execution result to select the optimal solution.
>
> - **ODEX** introduces the first open-domain execution-based natural language to Python code generation dataset. This dataset includes 945 pairs of natural language-code pairs, covering 79 different libraries, and provides 1,707 manually written test cases for execution validation.
>
> - **Self-OSS-Instruct** employs context learning techniques to enable the StarCoder2 model to self-generate diverse programming instructions from seed code snippets. This process includes concept extraction, where the model identifies key concepts in the code, and instruction generation, where it creates specific programming tasks based on these concepts.
>
> We will integrate and expand upon these points in the revised version and conduct proofreading.
>
> ---
>
> ### Comment2&3: More accurate to cite the use of rejection sampling and Typos.
>
> Thank you for your comments. We will revise the citation and correct the typos to improve the professionalism of the paper.

---

> > ### Author Response · Authors · 2024-11-23
> > **Response (2/2)**
> >
> > ### Q1: I am interested in counter-examples where programs fail to model the constraints in the instructions, as well as how to identify these cases and refer them to human experts.
> >
> > Thank you very much for your suggestions. The motivation of AUTOIF is to generalize more complex instruction-following abilities using an automatic scalable method through simple handwritten instructions with minimal human effort. Thus, the aim of this paper is for AutoIF to serve as an automated, plug-and-play solution widely applicable in the field of large model instruction following.
> >
> > However, we also believe that human expertise is essential for high-level semantic verification. Here are two areas where AutoIF can incorporate human intervention:
> >
> > 1. Human insights can be integrated during the seed instruction construction phase, allowing for intervention in enhancing the instructions while requiring only fewer than 100 handwritten instructions.
> > 2. Additionally, in the final stage of LLM verification, human-defined finer-grained principles can be incorporated into the verification prompt, and iterative modifications can be made to the prompts during small batch instruction verification.
> >
> > These are just some experiential thoughts, but they are not the main focus of this paper. In the future, we will also focus on optimizing human-in-the-loop strategies for instruction-following alignment to achieve broader alignment capabilities in large models.

---

### Official Review · Reviewer_61cd · 2024-11-04

**Soundness:** 3
**Presentation:** 4
**Contribution:** 2
**Rating:** 6
**Confidence:** 2

**Summary:**

This work presents AUTOIF, a generic and automated framework for the generation of high-quality training data to enhance the instruction-following prowess of LLMs. The main novelty of this work is that it reduces instruction-following alignment to a verifiable process using code. AUTOIF leverages execution feedback from code verification and rejection sampling to guarantee that the generated responses meet given constraints. This technique has paved the way for preparing both positive and negative examples for algorithms using SFT and RLHF. The authors mention that AUTOIF substantially improves the performance of LLMs on many benchmarks without weakening other competencies, such as coding and mathematical reasoning.

Besides this, the paper presents an extensive experiment to show that AUTOIF improves the instruction-following accuracy of a range of LLM architectures and sizes, from Qwen2 and LLaMA3 models. The paper underlines that indeed one of the keys to good performance is iterative online training and at the same time develops the scalability and generalization potential of AUTOIF across diverse types of instructions.

**Strengths:**

Extensive Performance Improvements: AUTOIF leads to significant gains across various instruction-following benchmarks, such as IFEval and FollowBench. It notably achieves over 90% loose instruction accuracy on IFEval while preserving or even slightly improving performance on other key capabilities like coding and mathematical reasoning.

Comprehensive Evaluation and Ablation Studies: The research includes a thorough evaluation of AUTOIF across different models, parameter sizes, and ablation settings, demonstrating the robustness and scalability of the approach. It also explores data scaling and efficiency, highlighting the method's potential to achieve strong performance with relatively fewer data samples.

Balanced Capability Improvement: AUTOIF enhances instruction-following without compromising other model abilities, such as coding and general reasoning, making it a well-rounded solution.

**Weaknesses:**

Conceptual Rigidity: While the core contribution of this paper is insightful, the attempt to cast the instruction-following alignment into a code verification problem introduces some rigidity. This is so because in that respect, focus will be on "verifiable instructions" that can be amenable to mechanical verification by code executors. While the structure for this argument is robust in this text, it might indeed be narrow to encompass more complex or subjective tasks. The paper does not delve into the non-code-based verification methods like model-based scoring or human-in-the-loop strategies; the thrust of the discussion remains centered on the execution feedback mechanism​.

Generalization Concerns: Concentration on verifiable instructions has been made amply clear in the method section. AUTOIF framework uses code for instruction verification, which in itself inherently limits the generalization to open-ended or subjective instructions. While the paper does highlight improvements across benchmarks such as IFEval, it does not really investigate how the approach generalizes when the tasks are less structured or ambiguous. Though the second concern with respect to AUTOIF biasing the models to easily verifiable instructions might be well-taken, given the current evidence, it should be framed as a hypothesis.

Somewhat worrying lack of methodological diversity: there is heavy focus on execution feedback. Other methods are not considered, like human feedback or model-based evaluation. Still, the paper does state that execution feedback and iterative optimization work fine, so there is a possibility that with more methods added, the instruction-following capabilities could be further improved.

It is very clear that, when explaining the strengths of the method, it seems over-dependent on automation and slighting the benefits that could be derived from purposive human intervention. The consideration of benefits from a more balanced approach in complex situations is thus little. By allowing human insights to complement this framework, therefore, it would adapt to inclusivity in the given context.

Novelty: Automatic instruction verification might not be an entirely new idea. Works in this area are CodeLlama2, PLUM, LLaMA3, SelfCodeAlign, and DeepSeek-Coder-V2. In fact, research efforts such as code-based execution feedback and automated instruction alignment have been made. For example, CodeLlama2 and LLaMA3 utilized the base of code-based synthesis and validation frameworks in their architectures for better performance on top of instruction-following tasks, which underlines code execution as a natural source of feedback. Simultaneously, these related works suggest that perhaps AUTOIF, with the novelty of scalability and data synthesis, is not really a new fundamental direction in the space.

**Questions:**

How could the framework be extended to manage instructions that are inherently ambiguous or have multiple valid responses?

What modifications to AUTOIF would be necessary to accommodate tasks involving emotional nuance or creative outputs?

Could providing more detailed information on errors and failure rates improve the replicability and robustness of AUTOIF?

What specific criteria were used to filter low-quality instructions, and can this process be made more transparent?

What are the computational overhead implications of using execution feedback at scale, and are there any optimizations or trade-offs that can be explored?

How does AUTOIF perform in resource-constrained environments, and is there a way to quantify these trade-offs more concretely?

---

> ### Author Response · Authors · 2024-11-20
> **Response (1/4)**
>
> Thanks for your helpful comments and we are glad that you acknowledged the strengths of our work! Below we will address your concerns point by point:
>
> ---
>
> ### W1(1) & W2 & Q1: AUTOIF will focus on "verifiable instructions," which may lead it to favor easy-to-verify tasks like IFEval, potentially excluding more complex or subjective tasks.
>
> Thank you for your insightful suggestion. First, AUTOIF not only incorporates Executor Feedback to automate the synthesis of verifiable instructions, but also introduces mechanisms of "LLM's Quality verification" and "Back-Translation Verification" to address more diverse and semantically complex instruction-following tasks. Therefore, AUTOIF relies not only on code execution results but also on the diverse scoring mechanism to validate generated responses from multiple perspectives, thereby improving data reliability and quality from semantic level. This significantly differs from the IFEval dataset, which focuses solely on rule-verifiable instructions. We hope the reviewers can appreciate the comprehensiveness of our innovations in verification. The ablation study (Table 4) also demonstrates that these mechanisms complement each other.
>
> Additionally, we would like to clarify that the datasets evaluated in Tables 1 and 2 encompass more complex and subjective tasks that reflect real-world scenarios, with difficulties far exceeding that of IFEval:
>
> - **FollowBench**: This assesses instruction-following across **over 50 integrated NLP tasks** with six difficulty levels. Most instructions cannot be validated and require GPT-4 for evaluation, as detailed in Appendix 6.
>
> - **MT-Bench**: A comprehensive evaluation set focusing on challenging questions to differentiate model capabilities, it includes 80 high-quality multi-turn dialogue questions across eight use cases: **writing, role-playing, information extraction, reasoning, mathematics, coding, STEM, andhumanities/social sciences**.
>
> - **InfoBench**: This dataset further validates our **out-of-distribution instruction-following capabilities**, also lacking verifiable instructions, with all evaluations requiring external model scoring.
>
> - **Arena Hard (win rate)**: One of the most challenging evaluation sets, aimed at **realistically assessing comprehensive chatbot performance**.
>
>
> Moreover, we have also included performance metrics on the long-text evaluation dataset FollowRAG [1].
>
>
>
>
>
> | Model                            | NQ (IF) | NQ (RAG) | NQ (AVG) | TQ (IF) | TQ (RAG) | TQ (AVG) |
> |----------------------------------|---------|----------|----------|---------|----------|----------|
> | Llama3-8B-SFT (Sharegpt+NQ+TQ)      | 15.7    | 59.5     | 37.6     | 15.0    | 76.5     | 45.7     |
> | Llama3-8B (AutoIF)              | 41.3    | 62.4     | 51.9     | 40.3    | 77.6     | 60.0     |
>
>
> We believe these integrated tasks effectively validate AUTOIF's generalization in real-world applications and hope these results can alleviate your concerns.
>
>
> ---
>
> ### W1(2): The paper does not delve into the non-code-based verification methods like model-based scoring or human-in-the-loop strategies
>
>
>
>
> Thanks for your comment! Regarding your concerns about the lack of non-code verification methods, in fact, as shown in Table 4 of the ablation study, the setting for ablation of cross verification (executor-based filtering) refers to the synthetic data of AutoIF, where we retain only the final stage of LLM query quality verification and do not conduct the executor feedback cross-validation process. This is essentially equivalent to the baseline process you mentioned, as in this setting, the data originally filtered by code execution will ultimately undergo LLM-based query quality verification. Below are the detailed experimental results:
>
>
>
> | Model                                                     | Pr. (L) | Ins. (L) | FollowBench (SSR) |
> |-----------------------------------------------------------|---------|----------|-------------------|
> | Qwen2-7B (LLM Verification only)                         | 44.6    | 52.9     | 52.9              |
> | Qwen2-7B (LLM Verification + Back Translation Verification) | 45.2    | 55.5     | 55.3              |
> | Qwen2-7B (AutoIF)                                       | 46.6    | 57.9     | 56.6              |
>
>
> It is clear that executor feedback plays a vital role in our AutoIF process, while model-based correction methods may be limited by insufficient self-knowledge and exhibit weaker performance. Although more model scoring strategies, such as human-in-the-loop approaches, exist, this paper focuses on automation and scalable enhancement of instruction following with minimal human effort. While this is not the primary focus of our work, we consider it an important area for future optimization and will address it in subsequent research.

---

> ### Author Response · Authors · 2024-11-20
> **Response (2/4)**
>
> ### W3: AutoIF is overly focused on executor feedback and does not consider other methods, such as human feedback or model-based evaluation.
>
>
> Thank you for your suggestion. We have already provided an response in W1 (1). In fact, AutoIF introduces a triple verification mechanism. In addition to incorporating "Executor Feedback" to automate the synthesis of verifiable instructions, it also introduces a dual mechanism of "LLM's Quality verification and Back-Translation Verification" to address more diverse and semantically complex instruction-following tasks. Therefore, AUTOIF relies not only on code execution results but also on the diverse scoring mechanism to validate generated responses from multiple perspectives, thereby improving data reliability and quality from semantic level. This significantly differs from the IFEval dataset, which focuses solely on rule-verifiable instructions. We hope the reviewers can appreciate the comprehensiveness of our innovations in verification. The ablation study (Table 4) also demonstrates that these mechanisms complement each other.
>
>
> ---
>
> ### W4: Allowing human insights to supplement the framework enables better adaptation.
>
> This is a good point! The motivation of AUTOIF is to generalize more complex instruction-following abilities using an automatic, scalable method through simple handwritten instructions with minimal human effort. Therefore, our intention is for AutoIF to serve as an automated, plug-and-play solution widely applicable in the field of large model instruction adherence, which is its advantage. To address your concerns, we propose two areas where AutoIF can incorporate human intervention:
>
> - Human insights can be involved in the seed instruction construction phase, allowing for intervention at the source to enhance instructions while only requiring fewer than 100 handwritten instructions.
> - In the final stage of LLM verification, we can integrate more granular principles established by humans into the verification prompt and iteratively modify the prompts during small batch instruction validation.
>
> These are merely some experiential thoughts and are not the main focus of this paper. We are open to addressing further style optimization in future work.
>
>
>
> ---
>
>
> ### W5: Automatic instruction verification is not novel, with related works emphasizing instruction alignment via code-based feedback. While AUTOIF introduces innovations in scalability and data synthesis, these studies suggest it does not represent a fundamentally new direction.
>
> Thank you for your suggestion. We would like to clarify the distinctions between our work and the Code LLM efforts you mentioned as follows:
>
> - **From the perspective of code generation goals**: The referenced works focus on code generation tasks aimed at producing more reliable code, while AutoIF's goal is to enhance the general instruction-following capabilities of LLMs under complex natural language instructions, with code used solely to verify whether the model's responses meet the criteria.
>
> - **From a technical standpoint**: Codellama2’s core innovations focus more on long context and code completion tasks. DeepSeek-Coder-V2 trains a reward model using a Python executor as the basis for subsequent RL optimization. In Llama3, the sections related to code training emphasize filtering overall code and checking for syntax errors in sub-units. These works are significantly different from AutoIF. Although SelfCodeAlign is somewhat similar, as it generates code for test case validation to filter high-quality responses, AutoIF emphasizes using program execution to validate the model's instruction-following ability, which is a cross-capability verification method.
>
> Furthermore, it is important to note that the related works focus on instruction tuning for code generation to enable general code alignment capabilities in LLMs, while AutoIF specifically aims to enhance LLMs' ability to follow complex and natural instructions, which are significantly different objectives.
>
> In addition, AutoIF introduces "executor feedback" along with "LLM-based quality verification scoring" and "back-translation verification mechanisms". Specifically, instruction-following tasks are more diverse and semantically rich, so AutoIF not only relies on code execution results but also utilizes a scoring mechanism to validate generated responses from multiple angles, enhancing data reliability and quality. We sincerely hope that reviewers will appreciate the comprehensiveness of our innovations in verification. The ablation study (Table 4) also demonstrates that these verification mechanisms complement each other.

---

> > ### Author Response · Authors · 2024-11-20
> > **Response (3/4)**
> >
> > ### Q2: What modifications to AUTOIF would be necessary to accommodate tasks involving emotional nuance or creative outputs?
> >
> >
> > We appreciate your valuable suggestion regarding integrating human interaction mentioned in W3 and W4. According to these, we believe AUTOIF can adapt to design emotional nuances and creative outputs in the following ways:
> >
> > - **Handwritten prompts**: We can consider fine-grained emotional differences in the prompts by handwriting instructions that allow for nuanced distinctions.
> > - **Instruction rewriting phase**: We can establish creative principles (e.g., for an emotional assistant, qualities like humor and empathy) and allow humans to iteratively optimize these principles based on the quality of generated outputs from small batches, potentially using instruction evolution techniques like AutoEval instructions [1].
> > - **Principle of LLM verification**: Inspired by CAI [2], we also need to incorporate fine-grained emotional differences in the verification prompts during the verification phase or use creative metrics for scoring, rather than solely focusing on instruction correctness, to overcome the limitations of executor-based verification that only addresses verifiable prompts.
> > - **Online/Offline DPO data construction**: For creative tasks, we should avoid using executor-based success rates to construct positive and negative samples. Instead, a combination of LLM verification scores and executor-based scores should be employed to balance correctness with higher-level emotional semantics.
> >
> > These are just some experiential thoughts and are not the main focus of this paper. We are open to addressing further optimization in future work.
> >
> >
> > ---
> >
> >
> > ### Q3: Could providing more detailed information on errors and failure rates improve the replicability and robustness of AUTOIF?
> >
> > Thank you for your insightful comment. In Table 5, we conducted a preliminary discussion on the relationship between large models' coding abilities and instruction-following capabilities, showing the pass rates for test cases. To enhance the completeness of our experiments, we incorporated your suggestion to perform a reliability analysis of generated code and testing. We sampled 10,000 instruction verification codes generated by GPT-4 and Qwen2 72B and analyzed their execution rates. After excluding external errors (such as compiler environment issues), we will present and analyze the pass rate results alongside Table 5.
> >
> > | Model      | Code Execution Rate | Case Pass Rate | MBPP | IFEval |
> > |------------|---------------------|----------------|------|--------|
> > | GPT-4     | 76%                 | 28%            | 73.9 | 59.3   |
> > | Qwen2-72B | 68%                 | 34%            | 87.5 | 44.7   |
> >
> > The experiments show that the code execution rate is reasonable and positively correlated with the code capabilities of both models. Notably, MBPP performance, code pass rate, and instruction-following capabilities are consistently positively correlated. This reinforces our conclusion on "Data Efficiency," where the supervision model's coding ability significantly impacts data synthesis quality and final instruction-following capability.
> >
> > ---
> >
> > ### Q4: What specific criteria were used to filter low-quality instructions, and can this process be made more transparent?
> >
> >
> > Thank you for your suggestion. To enhance the reproducibility of AUTOIF, the filtering criteria are completely transparent.
> >
> > 1. **Executor-based verification**: In lines 211-234, we detail the four standards used for our cross-validation.
> > 2. **Back-translation verification**: As mentioned in lines 246-249, we recall the instruction and predict using an NLI model, stating, "We filter out any instruction I labeled as a contradiction to ensure intent consistency."
> > 3. **Instruction-following verification**: Lines 265-269 also present detailed criteria.
> > 4. **Query quality verification**: In lines 275-276, we describe scoring from 1 to 10 and filter out samples with a score lower than 8.
> >
> > For more detailed prompt information and execution processes, please refer to the appendix and supplementary materials, where we provide comprehensive descriptions.
> >
> >
> > ---
> >
> > ### Q5: What are the computational overhead implications of using execution feedback at scale, and are there any optimizations or trade-offs that can be explored?
> >
> > As we provided detailed code in the supplementary materials, all our compilation feedback can be completed in bulk on a CPU with a single click. To optimize compilation speed, we can use multiprocessing; Python's `multiprocessing` package makes this process easy to implement. We believe this overhead is significantly lower than using large models for verification.

---

> > > ### Author Response · Authors · 2024-11-20
> > > **Response (4/4)**
> > >
> > > ---
> > >
> > > ### Q6: How does AUTOIF perform in resource-constrained environments, and is there a way to quantify these trade-offs more concretely?
> > >
> > >
> > > Regarding concerns about other computational resources, such as GPUs and the GPT-4 API, we report in Table 1 that we are using deployable open-source models like Llama3-70B and Qwen2-70B as supervision models, rather than GPT-4. To further alleviate your concerns, we present a setup using Llama3-8B for self-alignment and enhancing Qwen2-7B with Qwen2-3B-instruct. These supervision models can be effectively deployed **using just one GPU**.
> > >
> > > |Method        | Pr. (strict) | Pr. (L) | Ins. (S) | Ins. (L) | FollowBench (AVG) |
> > > |----------------------------|---------------|---------|----------|----------|------------------|
> > > | **Supervision Model: Qwen2-3B-Instruct**      |               |         |          |          |                  |
> > > | Qwen2-7B-base              | 37.7          | 43.6    | 49.4     | 53.4     | 52.3             |
> > > | Qwen2-7B (ShareGPT)       | 30.9          | 33.5    | 42.4     | 45.2     | 38.1             |
> > > | Qwen2-7B (AutoIF)         | 40.3          | 46.0    | 53.5     | 56.8     | 53.0             |
> > > | **Supervision Model: Llama3-8B-Instruct**     |               |         |          |          |                  |
> > > | Llama3-8B-base             | 24.6          | 26.1    | 38.1     | 39.7     | 11.6             |
> > > | Llama3-8B (ShareGPT)      | 23.7          | 26.4    | 33.8     | 37.1     | 38.1             |
> > > | Llama3-8B (AutoIF)        | 32.5          | 37.7    | 43.3     | 49.2     | 44.2             |
> > >
> > >
> > > I hope the above results can alleviate your concerns about computational resources!
> > >
> > > ---
> > >
> > > ### Reference:
> > >
> > > [1] Automatic Instruction Evolving for Large Language Models
> > >
> > > [2] Constitutional AI: Harmlessness from AI Feedback

---

> > > > ### Author Response · Authors · 2024-11-25
> > > > **A Kind Reminder for Reading the Response**
> > > >
> > > > Dear Reviewer 61cd,
> > > >
> > > > Thank you for your insightful suggestions. We have done our best to address your concerns. Since the rebuttal period is closing very soon, could you please check the response to see whether it mitigates your concerns? We would greatly appreciate that!
> > > >
> > > > Thank you for your time and consideration, the authors.

---

### Author Response · Authors · 2024-11-26
**General Response to Reviewers and Revision Submitted**

We thank all the reviewers for their insightful comments and suggestions. We have revised the paper to address the reviewers’ concerns. Below we summarize the major revisions (the main revisions are marked with **red text** in the pdf, we also made some minor layout changes to fit the page limit), while we reply to the comments of each reviewer separately.


- We have included results from the long-text instruction-following evaluation set, FollowRAG [1], as detailed in the section "VALIDATION ON RAG SCENARIO." and Table8.

- We also added more AUTOIF configurations to ensure data synthesis can be executed under low-resource conditions (one GPU), particularly the Weak-to-Strong setup. See the section "MORE SETTINGS ON LOW RESOURCE SCENARIO" and Table9 for details.

- To validate the superiority of AUTOIF's online alignment, we included experimental comparisons with the OAIF [2] baseline alignment method, as discussed in the section "COMPARISON WITH OTHER RAILF BASELINE."  and Table 10.

- We expanded our discussion on related works in code execution to clarify the connection between AUTOIF and existing code execution efforts. Please refer to the section "DISCUSSION ON CODE EXECUTION WORKS."

- In response to the reviewers' expectations regarding the optimization direction of AUTOIF, we summarized this in the section "FUTURE WORK."

- We have also made consistent revisions regarding the citations, formulas, and typos mentioned by the reviewers.

We would like to express our gratitude to all the reviewers for their professional suggestions and patient discussions. Thank you!

---

**Reference**

[1] Toward General Instruction-Following Alignment for Retrieval-Augmented Generation

[2] Direct Language Model Alignment from Online AI Feedback

---

### Meta-Review · Area_Chair_BLp4 · 2024-12-21

**Metareview:**

This paper proposes AUTOIF, a Self-Instruct-based approach to synthetize instruction following data that can be programmatically verified using test cases (e.g., “generate three sentences all ended with letter e”). Experiments on SFT, DPO and an online DPO setting demonstrate that AUTOIF significantly improves the base model’s instruction following performance, without compromising general math and coding skills. Notably, AUTOIF is the first to surpass 90% accuracy on IFEVAL with loose instruction accuracy.

**Strengths:**

All reviewers agree that self-supervised methods for instruction following is an important research direction and the proposed method is intuitive and effective. The experiments and ablation studies are also quite comprehensive (61cd, 7mrPm, 5kXL), “covering a wide range of benchmarks” (BmDh). The proposed method can work with both SFT and DPO-based approaches (BmDh), without compromising other model abilities (61cd).

**Weakeness:**

Perhaps the most important issue of the approach is that it is limited to instruction following problems that can be verified using code (61cd, 5kXL), therefore it is unclear whether this approach would generalize to complex tasks with “semantically rich instructions” (5kXL), or tasks with “open-ended or subjective instructions” (61cd). There are also concerns about the novelty of this approach given the vast majority of related works on synthesizing coding data using test diagnostics (61cd, 7mrP, 5kXL). However, as the first paper that generalizes similar lines of research to general-purpose instruction following, this paper already makes significant contributions to the community. Therefore, the decision is Acceptance.

**Additional Comments On Reviewer Discussion:**

Most minor issues of the paper, such as additional ablations or evaluations on more general-purpose IF benchmarks, are addressed during the rebuttal phase. The reviewers also acknowledge the author's response regarding the novelty and the existence of related works in coding domain. There are no remaining open issues after the rebuttal phase.

---

Note: several first-time reviewers gave a score of 8. While I believe that the paper should be a clear accept, given some issues with writing and the fact that the underlying idea is largely based on existing Self-Instruct work in code generation, I feel it would be more suitable as a poster presentation. However, I don't mind if the paper is presented as a spotlight.

---

### Decision · Program_Chairs · 2025-01-22

Accept (Spotlight)